# Hierarchy between forelimb premotor and primary motor cortices and its manifestation in their firing patterns

Akiko Saiki-Ishikawa[†], Mark Agrios[†], Sajishnu Savya[†], Adam Forrest[†], Hannah Sroussi[†], Sarah Hsu, Diya Basrai, Feihong Xu, Andrew Miri*

Department of Neurobiology, Northwestern University, Evanston, United States

## eLife Assessment

This study provides **important** insights as to how interacting brain areas produce movement during the execution of a skilled multi-directional reaching task. Using a combination of single neuron and neural population analysis, optogenetic stimulation, and computational models, the authors provide **convincing** evidence of an asymmetrical influence between mouse premotor and motor cortex during the execution of a well-practiced behaviour. This asymmetry can only be captured by some but not all population analysis methods, which is a key lesson to the field in and of itself. Analyzing how activity that is shared and private to these areas relates to different aspects of movements, and why different methods provide different outcomes regarding the nature of inter-area interactions would further strengthen this work.

*For correspondence:
andrewmiri@northwestern.edu

[†]These authors contributed equally to this work

Competing interest: The authors declare that no competing interests exist.

## Abstract

Although hierarchy is commonly invoked in descriptions of motor cortical function, its presence and manifestation in firing patterns remain poorly resolved. Here, we use optogenetic inactivation to demonstrate that short-latency influence between forelimb premotor and primary motor cortices is asymmetric during reaching in mice, demonstrating a partial hierarchy between the endogenous activity in each region. Multi-region recordings revealed that some activity is captured by similar but delayed patterns where either region's activity leads, with premotor activity leading more. Yet firing in each region is dominated by patterns shared between regions and is equally predictive of firing in the other region at the single-neuron level. In dual-region network models fit to data, regions differed in their dependence on across-region input, rather than the amount of such input they received. Our results indicate that motor cortical hierarchy, while present, may not be exposed when inferring interactions between populations from firing patterns alone.

## Introduction

A hierarchical organization characterized by feedforward influence between neuronal populations is commonly imputed to neural systems (*Felleman and Van Essen, 1991*; *Bullmore and Sporns, 2009*; *Vezoli et al., 2021*). In the motor system, a range of observations spanning anatomy (*Campbell, 1904*; *Ueta et al., 2014*), lesion (*Passingham, 1985*; *Gremel and Costa, 2013*), activity (*Roland et al., 1980*; *Veuthey et al., 2020*), and activity perturbation *Schmidlin et al., 2008*; *Elliott et al., 2020* have been interpreted as reflective of a hierarchical organization in which premotor regions (PM) plan movements and exert their influence through primary motor cortex (M1), which executes them (*Scott, 2000*; *Fried et al., 2017*). Observation of substantial PM projections to brainstem and spinal cord (*Dum and Strick, 1991*; *Fisher et al., 2021*; *Hausmann et al., 2022*), bidirectional connection between PM and M1 (*Dum and Strick, 2005*; *Bundy et al., 2023*), and activity related to movement

planning in M1 and spinal cord (*Weinrich et al., 1984*; *Prut et al., 2001*) together inform an updated view of a partial hierarchy (*Fulton, 1935*; *Graziano, 2009*; *Morandell and Huber, 2017*). In this view, PM and M1 interact and each drive downstream motor circuits, though M1 exerts stronger influence on movement execution. Although 'hierarchy' has divergent meanings in neuroscience discourse (*Hilgetag and Goulas, 2020*), here we refer specifically to asymmetric reciprocal influence between two neuronal populations, with activity in one exerting a larger functional influence on activity in the other, in line with certain contemporary usages (*Morandell and Huber, 2017*; *Siegle et al., 2021*; *Merel et al., 2019*).

The empirical foundation for the view of a PM-M1 hierarchy remains incomplete. Such a hierarchy is realized when naturally occurring (endogenous) firing patterns in PM exert a larger influence on those in M1 than vice versa. However, existing observations have not yet resolved this asymmetric reciprocal influence between endogenous activity in PM and M1. Notions of PM-M1 hierarchy are instead based on observations like those of asymmetric effects on each region from electrical stimulation, lesion, or pharmacological inactivation of the other region (*Bucy, 1933*; *Stepniewska et al., 2014*); of differing laminar targets of projections between regions (*Rouiller et al., 1993*; *Hira et al., 2013*); and of differing degrees of activity in each region related to movement preparation (*Mushiake et al., 1991*; *Umilta et al., 2007*; *Dixon et al., 2021*). For example, in anesthetized monkeys (*Cerri et al., 2003*) and rodents (*Deffeyes et al., 2015*), stimulation in PM alters the effects of stimulation in M1 on muscles. Other work has used analysis of simultaneously measured activity in PM and M1 to support claims of feedforward PM-to-M1 influence (*Veuthey et al., 2020*; *Makino et al., 2017*; *Terada et al., 2022*). However, these sorts of observations do not necessarily imply asymmetric reciprocal influence of endogenous activity in PM and M1; such influence is not resolved by lesion or pharmacological inactivation and need not match the influence of artificially induced activity or agree with activity correlations or anatomical connectivity. Moreover, other observations from electrical stimulation (*Bundy et al., 2023*) and simultaneous activity measurements in PM and M1 (*Truccolo et al., 2010*; *Kimura et al., 2017*) are consistent with relatively symmetric reciprocal influence between PM and M1. Recent modeling of such activity measurements also suggests that PM-M1 interactions could change across task phases (*D'Aleo et al., 2022*). It has also been proposed that differences in activity related to movement preparation could reflect involvement of each region in somewhat different aspects of movement (*Wiesendanger et al., 1987*; *Graziano, 2006*), rather than hierarchy.

Because it is unclear if and when asymmetric reciprocal influence between PM and M1 exists, it also remains unclear how firing patterns reflect such influence. It is frequently assumed that feedforward signal flow from one region to another can manifest as the appearance of an activity pattern in one region and later in the other at a latency reflecting axonal conduction and synaptic transmission (*Gokcen et al., 2022*; *Issa et al., 2018*; *Siegel et al., 2015*; *Schwiedrzik and Freiwald, 2017*). Yet movement-related activity patterns in PM and M1 are diverse (*Churchland and Shenoy, 2007*), and it remains to be seen how well activity in these regions can be captured by similar but temporally offset (delayed) activity patterns. Feedforward influence between regions is also thought to manifest in the ability to predict firing patterns in one region from those in the other. The application of time series prediction methods for discerning functional influence between neurons (*Ito et al., 2011*; *Casile et al., 2021*) is grounded in the notion that feedforward influence leads to predictive relationships between firing patterns. We might expect that a partial hierarchy between PM and M1 would manifest as an asymmetry in firing pattern predictivity, where PM activity patterns share a stronger predictive relationship with subsequent M1 activity patterns than vice versa. However, local circuit dynamics can also be expected to play a substantial role in determining firing patterns. Recent theoretical results suggest that local circuit dynamics can be the primary determinant of firing patterns, even in the presence of substantial across-region input like that between motor cortical regions (*Bachschmid-Romano et al., 2023*; *Gozel and Doiron, 2023*). Thus, whether functional hierarchy manifests in firing pattern predictivity at the scale of brain regions within neural systems remains unresolved.

We sought to establish (1) hierarchy between PM and M1 mediated by their endogenous firing patterns and (2) how any such hierarchy manifests in the firing patterns in each region. To address the former, during reaching in mice we examined interactions between the forelimb PM (rostral forelimb area, RFA) and M1 (caudal forelimb area, CFA). To assess interactions between endogenous activity patterns on the fast timescales of synaptic communication between regions (oligosynaptic timescales), we used rapid optogenetic inactivation of either cortical region while simultaneously recording

with large-scale multielectrode arrays (Neuropixels *Jun et al., 2017*) in the other region. We found that influence between regions is asymmetric, indicating a partial functional hierarchy. To address (2), we then recorded simultaneously with Neuropixels in both regions and performed a range of analyses on measured firing patterns. Consistent with partial hierarchy, we found that some activity can be captured by similar but delayed patterns in which either region's activity led, with RFA activity leading more often. However, we also found a high degree of similarity between firing patterns in both regions and that firing patterns in each region had similar predictive relationships with subsequent firing in the other region at the single-neuron level. In dual-region network models fit to our activity measurements with and without optogenetic inactivation, regions differed in their dependence on across-region input, rather than the amount of such input they received. These results suggest that functional hierarchy, while present among endogenous activity patterns in RFA and CFA during reaching, is not reflected in firing patterns as often assumed. This has important implications for contemporary attempts to infer interactions between populations from activity patterns alone.

## Results

### RFA and CFA activity during directional reaching

We first trained mice to perform a voluntary limb movement task that we expect to depend on activity in both RFA and CFA. Building on recent work (*Galiñanes et al., 2018*), we developed a head-fixed directional reaching paradigm in which mice learn to rest their right hand on a rung and then reach to one of four spouts to grab a water droplet (*Figure 1A and B*; *Videos 1 and 2*). Rung touch illuminates a visual cue indicating the spout where the droplet will be dispensed after a subsequent (1–3 s later) auditory 'Go' cue. If mice moved their hand from the rung before the Go cue, the trial was aborted. The Go cue marked the start of a one-second response period, after which suction removed any uncollected water from the spout. Mice were acclimated to head fixation and trained to perform the task using methods similar to those previously described, a process which generally took between 10 and 20 daily sessions. To measure motor output during reaching, electromyographic (EMG) electrodes were chronically implanted into four muscle groups in the right forelimb: elbow flexors and extensors, and wrist flexors and extensors (*Miri et al., 2017*; *Figure 1C*).

We next aimed to show that RFA and CFA output influence muscle activity at oligosynaptic latency during this directional water reaching. In trained VGAT-ChR2-EYFP mice (*Zhao et al., 2011*) performing the reach task, we briefly projected a small spot of blue light (50ms, 1.5 mm diameter, 9 mW/mm$^2$) onto the surface of left RFA or CFA, triggered on reach onset. This approach silences the vast majority of endogenous activity in projection neurons at least down through layer 5 (*Miri et al., 2017*; *Guo et al., 2014b*), including those projecting to the brainstem and spinal cord; we show below that direct optogenetic effects were only seen in the targeted forelimb area and not the other. Activity was simultaneously recorded from right forelimb muscles to measure changes in motor output. For inactivations of either region, the absolute difference in activity summed across all muscles deviated from 0 very quickly after light onset (*Figure 1D and E*; *Figure 1—figure supplement 1*). This absolute difference was significantly different from zero when averaged over the first 25, 50, and 100ms after light onset (*Figure 1F*; for RFA, p=9.6 x 10$^{-8}$ for 25ms, p=1.3 x 10$^{-10}$ for 50ms, and p<10$^{-10}$ for 100ms; for CFA, p=3.0 x 10$^{-5}$ for 25ms, p<10$^{-10}$ for 50ms, and p<10$^{-10}$ for 100ms; one-tailed t-tests). This indicates that both RFA and CFA exert oligosynaptic influence on muscles during reaching.

We next examined whether neuronal firing patterns in the premotor RFA and primary motor CFA showed a commonly observed feature in primates and rodents alike (*Veuthey et al., 2020*; *Weinrich et al., 1984*; *Makino et al., 2017*; *Tanji and Kurata, 1982*): earlier activity preceding movement in the PM region. We acquired simultaneous large-scale multielectrode array (Neuropixel) recordings across cortical layers in both CFA and RFA in mice as they performed our reaching task (21 sessions across six mice, 74–293 qualifying single units per session in CFA, median 150; 17–194 per session in RFA, median 105; *Figure 2A–C*). As expected, the majority of recorded neurons in each region exhibited an elevated average firing rate during movement as compared to periods when forelimb muscles were quiescent, both for narrow-waveform, putative interneurons and wide-waveform, putative pyramidal neurons (*Figure 2D and E*; *Figure 2—figure supplement 1A and B*). The firing rate time series of a large fraction of neurons in each region was significantly correlated with the activity time series of at least one forelimb muscle (*Figure 2F*). For all subsequent analysis of simultaneous RFA and CFA

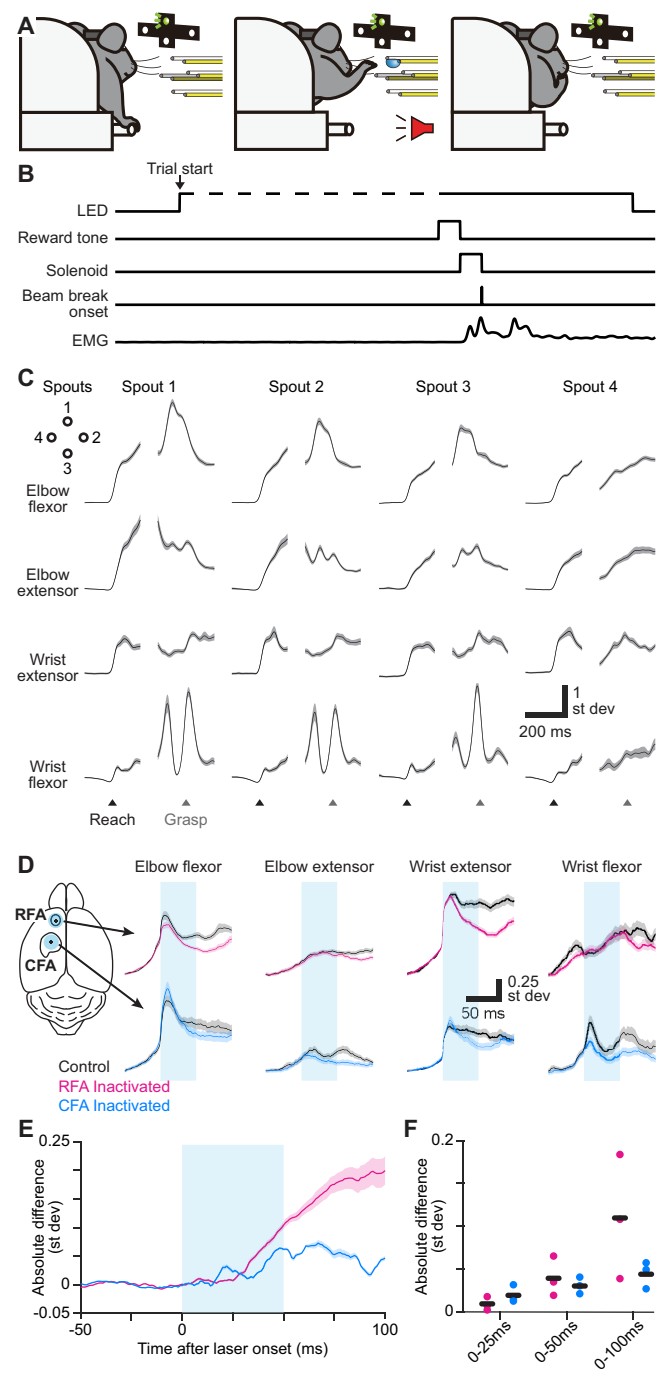

**Figure 1.** RFA and CFA influence on forelimb muscles during directional reaching in mice. (**A**) Schematic depicting the directional reaching task. (**B**) Schematic depicting the time course of experimental control signals and muscle activity during the directional reaching task. (**C**) Trial-averaged activity of four recorded muscles (black) ± SEM (gray, n=176 trials) for one mouse during directional reaching toward each of four spouts. Separate averages are aligned on reach onset (Reach) or spout contact (Grasp). Vertical scale bars in (**C**) and (**D**) reflect standard deviation of z-scored muscle activity. (**D**) For one example mouse, mean ± SEM muscle activity for trials without (black) or with inactivation (50ms, cyan bar) of RFA (top, magenta) or CFA (bottom, blue) triggered on reach onset. Left image shows the position of the light stimulus on RFA and CFA. (**E**) Mean ± SEM absolute difference between inactivation and control trial averages across all recorded muscles (n=12 from 3 animals) for inactivation (50ms, cyan bar) of RFA or CFA. For baseline subtraction, control trials were resampled to estimate the baseline difference expected by chance. This subtraction leads to values below zero. (**F**) Absolute difference between inactivation and

*Figure 1 continued on next page*

*Figure 1 continued*
control trials averaged over three epochs after light/trial onset, for individual animals (circles) and the mean across animals (black bars).

The online version of this article includes the following figure supplement(s) for figure 1:

**Figure supplement 1.** Extended time series from Figure F1D.

recordings reported here, to ensure reach behavior was similar across analyzed trials, we only included trials where mice successfully reached to the correct spout. We also excluded outlying trials in which the reach duration, reaction time, or baseline muscle activity was more than three standard deviations above the mean for successful trials, as well as trials in which the initial rate of change in muscle activity was more than three standard deviations below the mean for successful trials (see Methods).

We then quantified the activity change around reaching onset in each region. We used the absolute change in trial-averaged neural activity state from a pre-reach baseline summed over top principal components (PCs) that together captured >95% of the variance (2–3 PCs; *Figure 2—figure supplement 1C and D*). We measured the time at which the activity state deviated from baseline preceding reach onset, which occurred earlier in RFA than in CFA (*Figure 2G and H*; p=0.031, n=6 mice, one-tailed signed rank test); the mean time for RFA was 53.9ms earlier than in CFA. To ensure differences in onset were not the result of differences in baseline variance, we repeated the analysis using the same absolute threshold for each region. Here, we used the higher of the two thresholds from each region. We found that RFA onset still preceded CFA onset (p=0.031, n=6 mice), but the timing difference was reduced, with RFA activation preceding CFA activation by 28.8ms on average. We also performed a similar analysis on the trial-averaged firing rates of individual neurons. To better estimate onset times, we focused on neurons with mean firing rates above the 90th percentile value across all sessions within each animal. We found that the mean activity onset time for RFA neurons preceded that for CFA neurons (*Figure 2—figure supplement 1E and F*; p=0.031, n=6 mice). Finally, we found that the fraction of activity variance preceding movement onset was significantly higher for RFA than CFA (*Figure 2I*, p=0.016, n=6 mice). Thus, in our reaching paradigm, RFA exhibits earlier pre-movement activity compared to CFA.

## Asymmetric reciprocal influence of endogenous RFA and CFA activity

We then sought to establish that the endogenous activity in RFA exerts a larger influence on CFA activity than vice versa. In trained VGAT-ChR2-EYFP mice performing the reach task, we inactivated RFA as described above while recording activity across layers with Neuropixels in CFA (*Figure 3A-C*, 13 sessions across three mice), or similarly inactivated CFA while recording in RFA (*Figure 3D-F*, 10 sessions across three mice). Blue light pulses were again triggered on reach onset. This will inactivate similarly sized cortical regions that encompass all of RFA or most of CFA. Thus, our approach provides

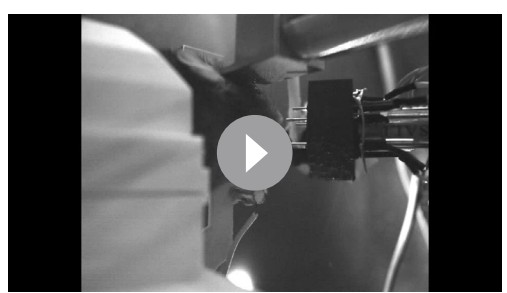

**Video 1.** Water reaching, side view. A side view of a mouse completing a trial of the water reaching task. LEDs are hidden behind the water ports from this view angle.

https://elifesciences.org/articles/103069/figures#video1

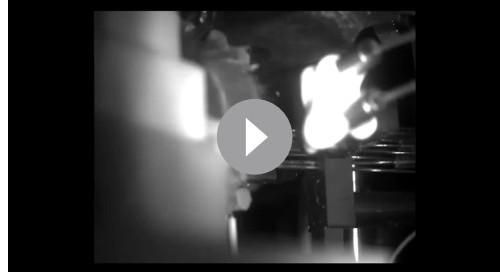

**Video 2.** Water reaching, rear view. A rear view of a mouse completing three trials of the water reaching task. LEDs can be seen in the middle of the screen. Graphics indicate when the LED, Go cue tone ('Cue'), and water dispensation ('Reward') occur. Rewards are achieved when the mouse maintains its hand on the rung for the duration of the rest period.

https://elifesciences.org/articles/103069/figures#video2

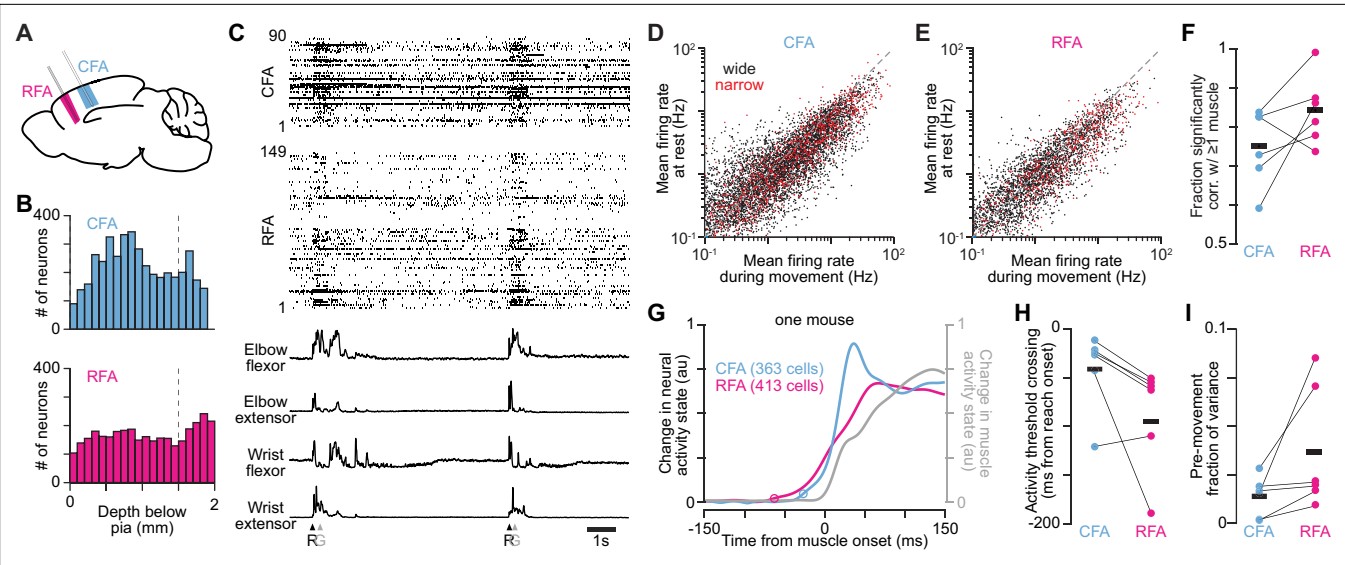

**Figure 2.** RFA and CFA activity during directional reaching in mice. (**A**) Mouse brain schematic depicting recordings in RFA and CFA. (**B**) Histograms showing the depth below pia of single unit isolations (neurons) recorded with Neuropixels and extracted with Kilosort. Dotted vertical lines depict where the cutoff for the bottom of each cortical region was defined. (**C**) Example of spike rasters (top) and muscle activity (bottom) recorded in one mouse during reach performance. Arrowheads indicate each onset (**R**) or spout contact (**G**). (**D**), (**E**) Scatter plots of the firing rates for neurons recorded in CFA (**D**) and RFA (**E**) during epochs when mice are activating their forelimb muscles, versus periods when all muscles are quiescent, separated for neurons that have wide and narrow waveforms. (**F**) The fractions of neurons recorded in both CFA and RFA whose firing rate time series was significantly correlated with that of at least one muscle (p-value threshold = 0.05), for individual animals (circles) and the mean across animals (black bars, n=6 mice). (**G**) For one mouse, normalized absolute activity change from baseline summed across the top three principal components (PCs) for all recorded RFA or CFA neurons, and the top PC for muscle activity. Circles indicate the time of detected activity onset. The pre-reach baseline epoch was from 150 ms to 100 ms before reach onset. Neural activity onset was detected as the first time at which activity rose 11 standard deviations above baseline. (**H**) Time from reach onset at which the activity change from baseline summed across the top PCs for RFA (magenta) or CFA (blue) neurons rose above a low threshold, for individual animals (circles) and the mean across animals (black bars, n=6 mice). (**I**) The activity variance in the 150ms before muscle activity onset, defined as a fraction of the total activity variance from 150ms before to 150ms after muscle activity onset, for each animal (circles) and the mean across animals (black bars, n=6 mice).

The online version of this article includes the following figure supplement(s) for figure 2:

**Figure supplement 1.** RFA and CFA activity during directional reaching in mice.

a relatively comprehensive inactivation, enabling measurement of the influence each region has on neuronal firing in the other region.

We sought to verify that ChR2 activation in VGAT[+] neurons was negligible in the region being recorded but not directly inactivated. We modified the Stimulus-Associated spike Latency Test (SALT *Kvitsiani et al., 2013*) to test for significant effects of light on spike latency in the first 5ms following light onset, specifically in narrow-waveform, putative inhibitory interneurons in the recorded region (*Figure 3—figure supplement 1A and B*). For each neuron, this yielded a p-value reflecting the likelihood that spike latency differences were due to chance (*Figure 3—figure supplement 1C–F*). The distribution of these p-values across all recording sessions was not significantly different from uniform, both for recordings in RFA and in CFA during inactivation of the other region (RFA: p=0.22, n=328 neurons; CFA: p=0.13, n=207 neurons; K-S test). We also observed that the effects on narrow-waveform neuron firing rates were similar in magnitude and time course to effects on wide-waveform, predominantly pyramidal neuron firing rates (*Figure 3—figure supplement 1G–R*). Thus, we were not able to detect direct ChR2 activation effects on narrow-waveform neurons in the recorded region. We also verified that inactivation effects on firing in the downstream region were stable both within and across sessions.

Results indicated an asymmetry in the oligosynaptic functional influence between CFA and RFA. Control and inactivation trial averages showed a larger reduction in CFA firing upon RFA inactivation compared to the effect on RFA firing upon CFA inactivation (*Figure 3A–F*). To quantify the effect on firing in individual neurons and to account for the possibility of both increases or

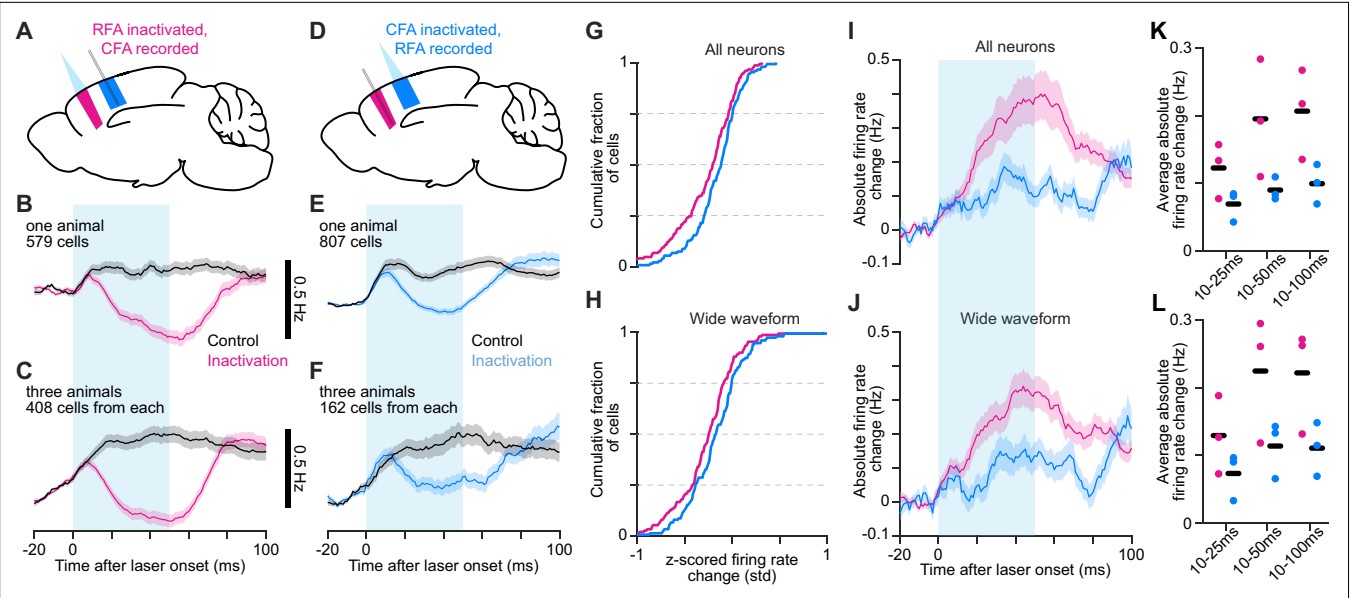

**Figure 3.** Asymmetric reciprocal influence of endogenous RFA and CFA activity. (**A–F**) Schematics depicting inactivation and Neuropixel recording (**A,D**) and mean ± SEM firing rate time series for neurons from one animal (**B,E**) or all three animals (**C,F**) from inactivating RFA and recording CFA (**A–C**), or vice versa (**D–F**). The cyan bar indicates light on. The same number of cells was used from each animal in (**C**) and (**F**). The minimum firing rates for the inactivation trial averages are as follows: (**B**) 0.72 Hz, (**C**) 1.18 Hz, (**E**) 0.42 Hz, (**F**) 0.89 Hz. (**G**), (**H**) Cumulative histograms of the difference between averaged z-scored firing rates for control and inactivation trials averaged from 45 to 55ms after light/trial onset, for the top 50 highest firing rate neurons (**G**) or wide-waveform neurons (**H**) from each animal, combined. (**I**), (**J**) Mean absolute firing rate difference ± SEM between control and inactivation trial averages for all (**I**) and wide-waveform (**J**) neurons recorded in the other area during RFA and CFA inactivation. The same number of cells was used from each animal. Baseline subtraction enables negative values. (**K**), (**L**) Average absolute firing rate difference from 10ms after light onset to 25, 50, and 100ms after for all (**K**) and wide-waveform (**L**) neurons, for individual animals (circles) and the mean across animals (black bars).

The online version of this article includes the following figure supplement(s) for figure 3:

**Figure supplement 1.** Effects of CFA and RFA inactivation on firing rates in the other region.

decreases in their firing rates, we computed the difference of control and inactivation trial averages 50ms after light/trial onset using z-scored firing rates for each neuron. To focus on the neurons with the best firing rate estimates, we used the neurons with the 50 highest average firing rates in each animal, either among all neurons or all wide-waveform, putative pyramidal neurons. Distributions of the relative firing rate changes for these neurons showed that the vast majority of cells showed decreases in firing rate, and increases were rare, both for all neurons (*Figure 3G*) and for wide-waveform neurons (*Figure 3H*). These relative firing rate changes showed the substantial magnitude of reductions for many neurons; substantial fractions of neurons showed reductions that were more than half the standard deviation of their firing rate over the recording session. We note here that because we have focused on high firing rate neurons, we cannot distinguish whether the strong bias toward decreases in firing rate is unique to such neurons, though we see no reason to expect it to be.

Examining the average absolute change in firing over time, we again found a substantially larger effect for CFA neurons than RFA neurons, both for all neurons (*Figure 3I*) and the wide-waveform subset (*Figure 3J*). The average absolute change in firing was significantly larger whether calculated over the first 50 or 100ms following light/trial onset, though it did not reach significance over the first 25ms (*Figure 3K and L*; all neurons: 25 ms p=0.077, 50 ms p=0.012, 100 ms p=0.021; wide-waveform: 25 ms p=0.101, 50 ms p=0.027, 100 ms p=0.022; one-tailed t-test). Thus, it is not the case that the smaller effect of CFA inactivation on RFA activity is solely due to offsetting increases and decreases in the firing rates of different neurons. These results demonstrate that the endogenous activity in RFA and that in CFA exert asymmetric effects on one another, with RFA activity exerting a comparatively larger effect on CFA activity. We interpret this as a direct indication of functional hierarchy on oligosynaptic timescales.

## Premotor and primary motor cortical activity are highly similar during reaching

We then examined how the asymmetric reciprocal influence between CFA and RFA manifests in the relationship between their firing patterns. Previous measurements with calcium sensor imaging (*Terada et al., 2018*) or non-simultaneous array recordings *Saiki et al., 2014* have shown substantial similarity between CFA and RFA activity during forelimb tasks. We therefore sought to quantitatively assess the degree of similarity at high temporal resolution from our simultaneous recordings in both regions during reaching. We used methods that decompose two sets of variables into pairs of components (linear combinations of the original variables) that are highly similar. We used both canonical correlation analysis (CCA *Hotelling, 1936*; *Sussillo et al., 2015*), which maximizes the correlation between the time courses for pairs of components, and partial least squares (PLS *Le Floch et al., 2012*), which maximizes the covariance of these time courses. Both methods were applied to matrices comprising the trial-averaged firing rates for all neurons from a given animal (*Figure 4A* and *Figure 4—figure supplement 1A*). In these matrices, the separate averages aligned on reach onset and on grasp for reaches to each of the four spouts were concatenated. This analysis was performed separately for each of six mice using all qualifying single units aggregated across sessions.

CCA revealed that the vast majority of the trial-averaged activity variance in RFA and CFA is captured by components with nearly identical time series. On average, over 80% of activity in both regions is captured by components whose time series had Pearson correlations >0.94 (*Figure 4B–D*). We compared the cumulative activity variance captured by successive canonical variables with that captured by equivalent numbers of principal components (i.e. the maximum of any possible components). On average, the first ten canonical variables captured 92.2 ± 2.4% (mean ± SEM) as much as the first ten principal components for CFA, and 92.6 ± 1.5% as much for RFA. PLS identified components that successively capture nearly the same amount of activity variance as corresponding principal components while still maximizing the covariance of component time series (*Figure 4E and F*). The first ten PLS components captured 98.7 ± 0.3% as much as the first ten principal components for CFA, and 98.2 ± 0.5% for RFA. To establish a baseline level of similarity expected by chance for both methods, we repeated calculations after replacing the firing rate time series segments used to generate trials averages with separate, randomly chosen segments for one of the two matrices. Similarity was markedly reduced in both cases (*Figure 4B–F*). Thus, both methods produced a similar basic result: much of the activity variance in both regions is parsimoniously explained by components that are very similar. This implies that trial-averaged firing patterns in RFA and CFA en masse are highly aligned. This similarity in firing patterns, coupled with the presence of descending projection neurons in both regions (*Wang et al., 2018*), suggests redundant and collective influence on motor behavior. Because our results here involve the vast majority of trial-averaged activity variance, we expect that they encompass both components of activity that vary for different movement conditions (condition-dependent) and those that do not (condition-invariant; *Kaufman et al., 2016*).

In a functional hierarchy where one region's activity rises earlier than another's, we might expect the alignment between sets of firing patterns to be maximal when one set is lagged in time relative to the other. We thus assessed whether there was a time lag greater than 0 at which RFA and CFA activity patterns during reaching would best align. We found this was not the case. We shifted CFA activity in time relative to RFA activity by each integer value from –30 to 30ms and recomputed CCA and PLS as above. In all cases, alignment quality was nearly identical: the average Pearson correlation for canonical variables was nearly constant for all lags tested (*Figure 4G*), as was the summed covariance of PLS component pairs (*Figure 4—figure supplement 1B*). However, control analyses in which we aligned CFA activity to itself, or to itself shifted by 10ms in time, revealed clear maxima in alignment quality at the expected lags (*Figure 4—figure supplement 1C and D*). Thus, the asymmetric reciprocal influence between RFA and CFA and the earlier activity change in RFA do not imply a maximal alignment at a lag.

## An imbalance in activity patterns shared at a lag

In a functional hierarchy, we might expect there to be activity patterns in the region exerting the larger across-region influence that then appear later in the second region because of synaptic time delays. We thus used a newly developed dimensionality reduction method, Delayed Latents Across Groups (DLAG *Gokcen et al., 2022*), that is designed to identify such patterns. We used DLAG

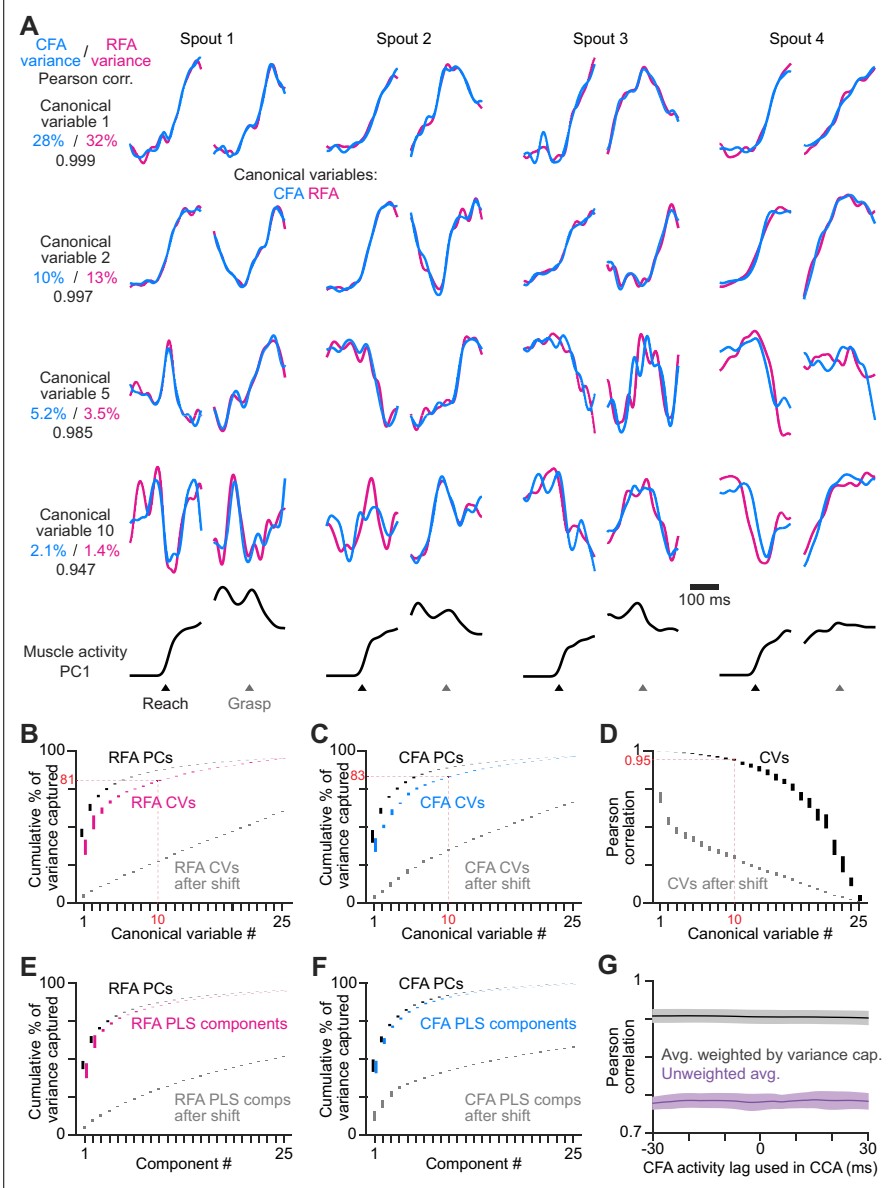

**Figure 4.** RFA and CFA firing patterns during reaching are highly similar. (**A**) From one animal, four canonical variables aligned at reach onset or spout contact (grasp) for reaches to each spout. Bottom row shows the corresponding time series for the first principal component of muscle activity. (**B**), (**C**) Mean ± SEM cumulative variance capture (n=21 sessions across 6 mice) for canonical variables (CVs, color), principal components (PCs, black), and CVs using shifted firing rate time series segments for one region as a control (gray), for RFA (**B**) and CFA (**C**) activity. Red annotations facilitate comparisons across B-D. Results in B-F all reflect n=21 sessions across 6 mice. (**D**) Mean ± SEM Pearson correlation for canonical variable pairs. (**E**), (**F**) Mean ± SEM cumulative variance capture for PLS components (color), principal components (PCs, black), and PLS components using shifted firing rate time series segments for one region as a control (gray), for RFA (**E**) and CFA (**F**) activity. (**G**) Mean ± SEM Pearson correlation of CFA and RFA canonical variable pairs computed when shifting CFA activity relative to RFA activity, averaged over all canonical variable pairs either without (purple) or with (black) weighting each correlation value by the average variance captured by its corresponding pair.

The online version of this article includes the following figure supplement(s) for figure 4:

**Figure supplement 1.** PLS alignment of RFA and CFA activity.

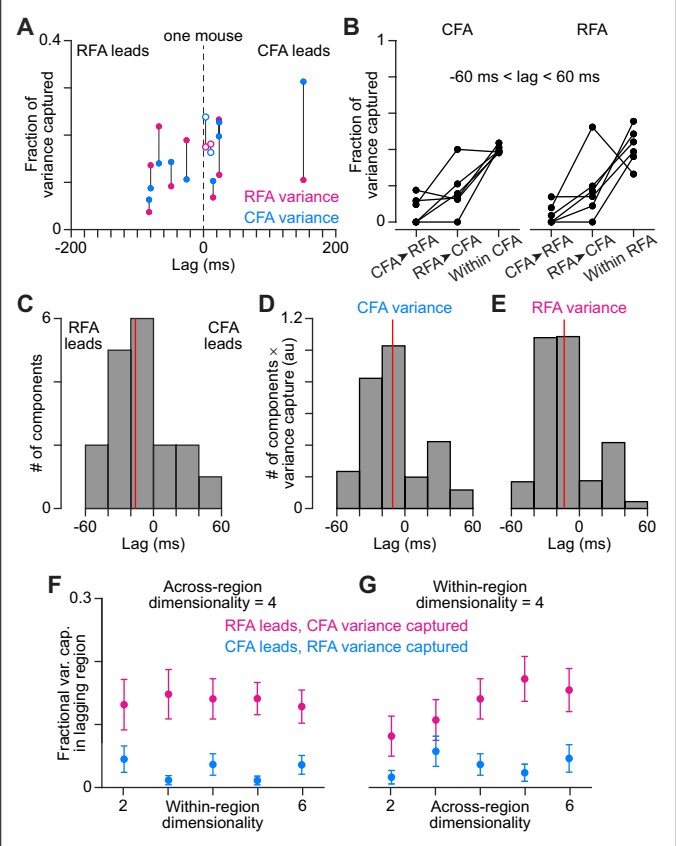

**Figure 5.** DLAG finds imbalance in variance capture by activity patterns shared at a lag (**A**). A scatter plot of the lags of all across-region activity components detected by DLAG versus their fractional variance capture in each region, for one mouse (3 sessions). Open circles reflect components where the lag was not significantly different from zero. Lines connect variance capture for individual components. Values for one component are not shown because its lag failed to converge between –200 and 200ms. (**B**) Fractional variance capture for CFA activity (left) and RFA activity (right) by across-region components in which CFA activity leads (CFA▶RFA) or RFA activity leads (RFA▶CFA), and by within-region components. Connected dots reflect the average across sessions for individual mice (n=6). (**C**) Histogram of DLAG component lags that were significantly different from zero and between –60 and 60ms, for all recording sessions (n=15). Red line indicates the median component lag. (**D**), (**E**) Histogram of DLAG component lags weighted by the variance each component captures in CFA (**D**) or RFA (**E**), for all recording sessions (n=15). Red lines indicate the mean component lag after weighting by variance capture. (**F**), (**G**) Mean ± SEM fractional activity variance captured in the lagging region by DLAG components when the lag was significantly different from zero and between –60 and 60ms, when varying the within-region (**F**) or the across-region (**G**) dimensionality (n=15 sessions).

The online version of this article includes the following figure supplement(s) for figure 5:

**Figure supplement 1.** Additional plots from DLAG calculations.

to decompose simultaneously recorded CFA and RFA firing patterns into latent variables (components) that are either unique to activity in each region (within-region) or shared between regions at an arbitrary temporal lag (across-region). For sessions of simultaneous CFA and RFA recording during reaching, we decomposed each region's activity into four across-region components (negative lags defined as RFA leading) and four within-region components.

*Figure 5A* shows representative results from one mouse, plotting the lag and variance capture in each region for all across-region components identified for three sessions. Some across-region components had a lag that was not significantly different from zero according to a permutation-based statistical test *Gokcen et al., 2022*; others had a lag that failed to converge between the boundary values of ±200ms. Both of these component types were ignored in subsequent analysis. Here, we focused the analysis on 15 sessions across the six mice, excluding sessions with a lower

number of trials, which hampered DLAG calculations. We also focused on across-region components with lags between –60 and 60ms, on par with the difference between the pre-movement activity rise time in RFA and CFA (see above, 53.9ms). We found that activity variance in each region could be captured by across-region components in which either CFA or RFA activity led with lags in this range (*Figure 5B*). This agrees with the breadth of results cited above that suggest bidirectional interaction between PM and M1.

We then examined whether there was any significant bias toward across-region components in which RFA activity led. For across-region components with lags between –60 and 60ms, we found such a bias (*Figure 5C*). The median lag for all across-region components was –16ms. The median of median lags for individual sessions was significantly less than 0 (p=0.021, one-tailed Wilcoxon signed-rank test). The same bias toward negative lags remained when including all components that had lags between –200 and 200ms (*Figure 5—figure supplement 1A*). We considered whether this bias remained when accounting for how much variance each component captured in either region. It did, as the bias remained prominent when we weighted components by their variance capture in CFA (*Figure 5D*, *Figure 5—figure supplement 1B*; mean variance-weighted lag = –10.8ms) or RFA (*Figure 5E*, *Figure 5—figure supplement 1C*; mean variance-weighted lag = –13.4ms). Thus, across-region activity components in which RFA activity led were more prominent than those in which CFA led.

We then examined specifically whether, consistent with a functional hierarchy, across-region components in which RFA activity led captured more activity variance in CFA than CFA-led components captured in RFA. Variance capture in the lagging region by RFA-led and CFA-led components indicated this was the case. The same held when repeating this analysis but varying either the number of within-region (*Figure 5F*) or across-region (*Figure 5G*) components, while holding the other fixed at four. The same again held when considering all components that had lags between –200 and 200ms (*Figure 5—figure supplement 1D and E*). We note that the bias we observe here should not be due to CFA firing patterns being noisier, as firing rates were on average higher in CFA. Collectively, our results with DLAG reveal activity patterns shared at a lag between RFA and CFA, with a bias toward patterns in which RFA activity leads.

## Symmetric firing pattern predictivity

We then asked whether asymmetric reciprocal influence between CFA and RFA manifests in an imbalance in the ability to predict firing patterns in one region from those in the other at the single neuron level. Given the lack of consensus on how to assess this predictivity, for comprehensiveness we applied three metrics that each make different assumptions about how one firing pattern enables or improves prediction of another: point-process Granger causality (*Casile et al., 2021*), transfer entropy (*Ito et al., 2011*), and convergent cross mapping (CCM; *Sugihara et al., 2012*). Although these methods can be used to probe for potential causal influence between neurons, we stress that we are not making causal claims here and are focusing only on what these metrics directly compute: how the firing of one neuron enables or improves prediction of firing in another.

We computed each metric for pairs of neurons recorded simultaneously during reaching, one in each region, with one defined as source and the other as target. To focus attention on the timescale over which we expect oligosynaptic influence to manifest, we considered models that use source activity at different lags ranging up to 30ms preceding firing in the target neuron and chose the lag that maximized predictivity. To focus calculations on cell pairs for which statistical power was greatest, we took the product of the average firing rates for neurons in each pair and found the 10,000 pairs with the highest product. To avoid directional bias due to differing firing rates (CFA firing rates were slightly higher on average), we algorithmically adjusted this set of pairs to match the overall firing rate distributions for neurons used from each area (*Figure 6—figure supplement 1A and B*).

In calculations of point-process Granger causality, two models are fit to the firing of a target neuron: one using the past history of all neurons in the ensemble, and another that excludes the source neuron. The difference in fit quality quantifies the unique prediction improvement provided by the source neuron, and a p-value is assigned that indicates the likelihood of a similar improvement by chance. Here, the ensemble involved all CFA and RFA neurons from the given recording session that fell within the 10,000 pair set after the algorithmic adjustment. Models were fit to activity from a set of reach trials, each spanning 200ms before reach onset to 800ms after. The distributions of p-values

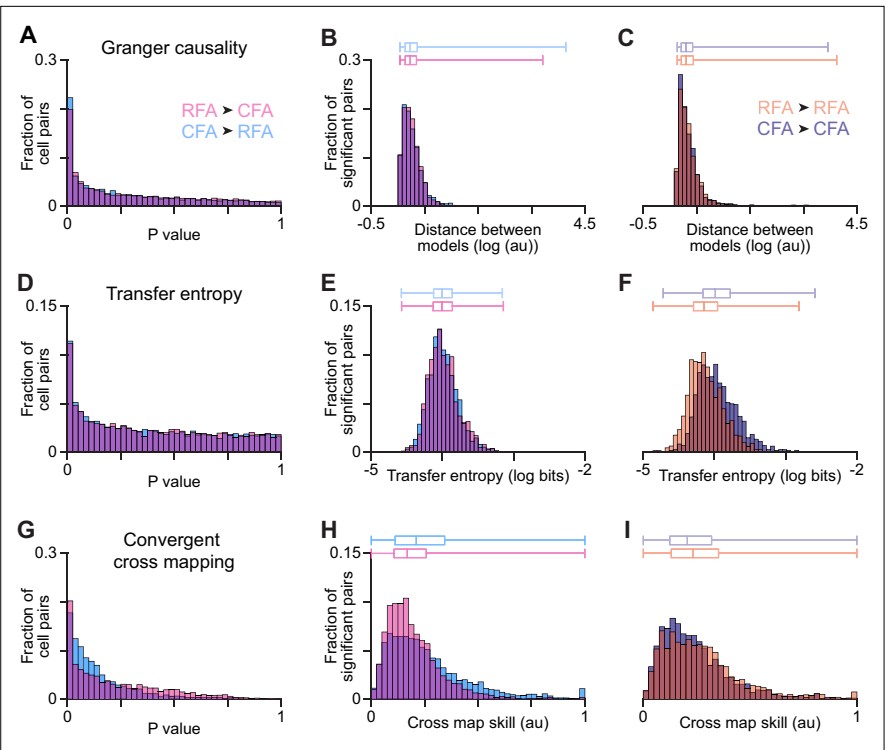

**Figure 6.** RFA and CFA firing pattern predictivity. (**A,D,G**) p-value distributions from calculations of Granger causality (**A**), transfer entropy (**D**), and convergent cross mapping (**G**) using firing patterns of across-region neuron pairs. (**B,E,H**) For across-region neuron pairs, distributions of metric values reflecting the improved prediction of target neuron firing using source neuron firing (**B and E**), or the improved prediction of source neuron firing using target neuron firing (**H**). Here and in (**C,F,I**), only values for which the corresponding p-value fell below a threshold (set to ensure the false discovery rate = 0.10) are included, and box plots on top show the minimum, 1st, 2nd, and 3rd quartile, and maximum values. (**C,F,I**) Distributions of metric values computed instead for within-region neuron pairs.

The online version of this article includes the following figure supplement(s) for figure 6:

**Figure supplement 1.** Additional plots from firing pattern predictivity calculations.

**Figure supplement 2.** Further neuron pair exclusion to avoid calculation anomalies.

for predictions in each direction (CFA source, RFA target, and vice versa) showed a skew toward zero and were significantly nonuniform (*Figure 6A*; KS tests, $p<10^{-10}$). This shows that for some fraction of pairs, source neuron activity did improve predictions of target neuron activity. However, the degree of skew toward zero was similar for both distributions, indicating that a similar fraction of pairs exhibited prediction improvements. We estimated the fraction of pairs showing prediction improvements from the p-value distributions (*Storey, 2002*), finding a slightly larger fraction when the CFA neuron was the source (CFA source: 69% of pairs, RFA source: 60% of pairs). Furthermore, the degree of prediction improvement in each direction was similar, as distributions of model fit quality differences for pairs with p-values below a significance threshold were similar (*Figure 6B*). Interestingly, these prediction improvements were similar to those computed instead for pairs of neurons within each region (*Figure 6C*), although there were more large extreme values within regions, potentially reflecting strongly coupled cells.

We found similar results with both transfer entropy and CCM. Transfer entropy measures how well the spiking of the target neuron can be predicted when considering its past history as well as the past history of the source neuron. The more the source neuron's past history improves the prediction of the target neuron's spiking activity compared to the target neuron's own past history alone, the higher the transfer entropy value. In this case, to increase our statistical power, rather than using the trial segments used for Granger causality calculations, we used spiking during all epochs of movement throughout a given recording session. To assign a p-value for each cell pair, we

generated an empirical null distribution by recomputing transfer entropy values after many different circular permutations of one neuron's spike time series for the given session. Here again, p-value distributions for predictions in each direction showed a skew toward zero and were significantly nonuniform (*Figure 6D*; KS tests, $p<10^{-10}$), evidencing prediction improvement for some pairs. The degree of skew toward zero here was very similar for both distributions. The estimated fraction of pairs showing prediction improvements was less than 1% different between distributions (CFA source: 29.4% of pairs, RFA source: 29.1% of pairs). The degree of prediction improvement was similar in each direction (*Figure 6E*) and was fairly similar to that computed for pairs within each region (*Figure 6F*).

CCM is designed to uncover causal interactions in a dynamical system with a weak to moderate deterministic component and where causal variables do not contain unique information. Here, we used it to quantify how well the activity of the source neuron can be predicted by the historical dynamics of the target neuron. Cross-mapping quality ('skill') was computed for 5000 high firing rate neuron pairs using a set of reach trials each spanning 200ms before reach onset to 800ms after. p-value distributions for predictions in each direction again showed a skew toward zero and were significantly nonuniform (*Figure 6G*; KS tests, $p<10^{-10}$), and the degree of skew toward zero was similar for both distributions. The estimated fraction of pairs showing prediction improvements was just 2.6% different between distributions (CFA source: 99.6% of pairs, RFA source: 97.0% of pairs). The degree of prediction improvement was similar in each direction (*Figure 6H*) and was fairly similar to that computed for pairs within each region (*Figure 6I*). Collectively, these results indicate that firing pattern predictivity computed for neuron pairs does not reflect an asymmetry like that seen in the actual influence of endogenous activity.

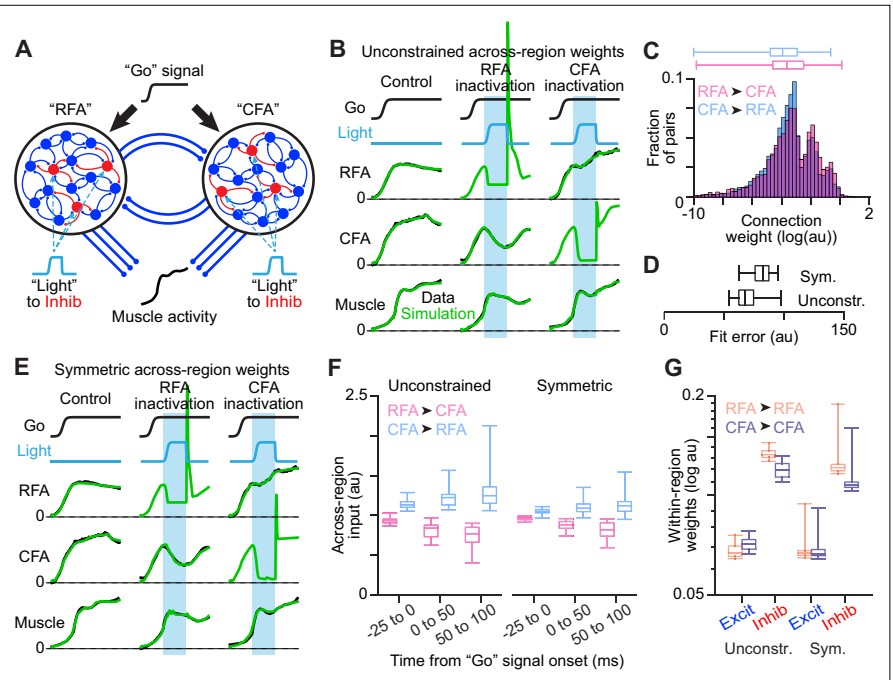

**Figure 7.** Network modeling of RFA and CFA activity. (**A**) Schematic of the dual network model fit to activity measurements. (**B**) Illustration of fit quality for one instance of the model with unconstrained weights of across-region synapses. In (**B**) and (**E**), cyan bars show the epochs of simulated inactivation. (**C**) Distribution of synaptic weights for all across-region connections from all instances of the unconstrained model. In (**C**), (**D**), and (**F**), box plots show the minimum, 1st, 2nd, and 3rd quartile, and maximum values. (**D**) Box plots for the distribution of error across 30 instances each of the unconstrained and constrained models. (**E**) Illustration of fit quality for one instance of the model with across-region weights constrained to be equal in each direction on average. (**F**) Distributions of summed across-region input (weights x activity) in each direction over different simulation epochs, for all instances of both the unconstrained and symmetric model types. (**G**) Distributions of within-region excitatory (Excit) and inhibitory (Inhib) synaptic weights, for all instances of both the unconstrained and symmetric model types.

## A network model recapitulates findings without asymmetric across-region input

Finally, we sought to develop intuition about how the sort of asymmetric reciprocal influence between CFA and RFA we observed could arise from two coupled neuronal populations. One possibility is that asymmetric influence could depend on asymmetric across-region input, with RFA sending stronger input to CFA than vice versa. On the other hand, asymmetric influence could instead arise from a difference in the dependence of activity in each region on that across-region input. In this scenario, differences between the local circuit dynamics in each region would cause a different degree of robustness to the loss of input from the other region.

We thus constructed a recurrent network model to observe how it would capture our results. We built a network of two populations ('RFA' and 'CFA'), each 80% excitatory and 20% inhibitory, with local recurrent (within-region) connectivity and sparser connectivity between populations (across-region) to align with experimental observations (*Figure 7A*). In response to a 'Go' signal, the network generates a muscle activity output. We trained instances of the model to generate measured muscle activity (summed across all four muscles) and measured neural activity (summed across all neurons recorded in a given region), in three cases: recording from either region while inactivating the other, and paired recording of both regions without inactivation. For the two inactivation cases, an additional input was provided to inhibitory interneurons to mimic ChR2 stimulation. Measurements for each case were drawn from different animals, and 30 model instances were computed from different random initial conditions. Both within-region and across-region connection weights were free to vary during training.

Analysis of fit results showed that networks do not capture the asymmetric influence we observed with greater input from 'RFA' to 'CFA'. Fit quality was very good, as measured neural and muscle activity were closely fit, and individual model instances readily exhibited both the temporal offset in activity rise and the substantial asymmetry in inactivation effects (*Figure 7B*). Models collectively had slightly larger 'RFA' to 'CFA' across-region weights (*Figure 7C*). However, repeating fits while constraining across-region weights to be balanced on average ('symmetric') yielded very good fits, with only a 14% average increase in fit error (*Figure 7D and E*). Most strikingly, the total amount of across-region input from each region, measured as the product of across-region weights and the activity of corresponding 'presynaptic' neurons, was not higher for 'RFA' to 'CFA' connections, both for unconstrained and symmetric models (*Figure 7F*). The across-region input was actually larger on average in the opposite direction, regardless of the time window over which the input was summed. We also observed that the 'RFA' network, which was more robust to the silencing of across-region input, had a higher average strength of within-region inhibitory connections (*Figure 7G*; unconstrained: $p = 1.9 \times 10^{-12}$, symmetric: $p = 8.2 \times 10^{-4}$, two-tailed t-test). These results indicate that asymmetric reciprocal influence can emerge in these networks from local recurrent connectivity and the activity dynamics they engender.

## Discussion

Here, we have explored the functional influence between the two main forelimb regions in mouse motor cortex and its manifestations in firing patterns during reaching. Using relatively comprehensive, optogenetic inactivation of either CFA or RFA while recording neural activity in the other, we found an asymmetry in the functional influence between regions, one aligned with existing views of a partial hierarchy between forelimb regions. Analysis of simultaneously recorded activity patterns in both regions detected an asymmetry in the presence of activity patterns shared at a lag. However, we also found that activity patterns in each region were in general very similar, and equally predictive of one another at the single-neuron level. Network models that capture population-level firing features and the asymmetric reciprocal influence between regions did so without an asymmetry in the amount of across-region input. Our results suggest that motor cortical hierarchy, while present in mice during reaching, is not reflected in motor cortical firing patterns as often assumed, and that firing patterns alone may not always clearly reflect the functional influence between regions.

### Functional hierarchy between forelimb regions

To properly interpret our activity perturbation results, it is important to consider how the effects of inactivation on connected regions may arise. Both CFA and RFA receive input from a broad range of other cortical regions, and each projects to a similarly broad range of regions (*Rouiller et al., 1993*;

*Oh et al., 2014*; *Urban Iii et al., 2024*). We expect that the most numerous inputs to neurons in each region are locally derived. Local recurrent circuits can give rise to activity patterns that evolve over timescales equivalent to many synaptic delays. When we silence either region, activity is thus disrupted in diverse synaptic pathways leading to neurons in the other region, including pathways through intervening regions. These disruptions will influence activity at different latencies, depending on conduction delays and the number of intervening synapses. By silencing either region, we can measure something approaching the full influence of the given region on the other, albeit one mediated through diverse synaptic paths. When assessing interactions between regions, we believe it can be important to account for this diversity of synaptic paths. Here, we observed a significantly larger influence of RFA on CFA than vice versa on all timescales examined, indicating that an asymmetry in reciprocal influence is present across timescales.

Our results highlight the dual determinants of inter-regional influence among brain regions: the balance of across-region input and the relative dependence of regions on that input. When we inactivate a particular region, the effects we observe in a downstream region will depend both on how much input it received from synaptic paths originating in the inactivated region and how robust firing patterns in the downstream region are to the loss of that input. Although classical descriptions emphasized the feedforward synaptic influence between regions in hierarchically arranged neural systems, observations of the relative prominence of local synaptic input (*Rossi et al., 2020*; *Thomson and Lamy, 2007*; *Braitenberg, 2014*) and contemporary recognition of the activity dynamics it can engender (*Ahmadian and Miller, 2021*; *Sadeh and Clopath, 2021*; *Seung, 1996*) indicate a relevance of robustness to input loss. Our knowledge of the relative robustness of networks in different cortical regions to input loss remains limited, but there are indications of variations in local recurrence that may induce variation in local network robustness (*Wang et al., 2019*; *Chaudhuri et al., 2015*; *Sanzeni et al., 2020*). Decoupling of functional influence between regions and their firing patterns like those we observed may be expected if local circuit dynamics are the primary determinant of inter-regional influence among brain regions.

These dual determinants of inter-regional influence should be kept in mind in interpreting the earlier pre-movement activity change we observed in RFA. Our findings thus call into question previous interpretations of earlier rising activity in PM as reflecting a hierarchy mediated by feedforward influence (*Veuthey et al., 2020*; *Weinrich et al., 1984*; *Umilta et al., 2007*; *Makino et al., 2017*). Although our results show RFA has a stronger direct influence on CFA than vice versa, this may not depend on the earlier rising activity in RFA, but instead on a differing robustness to input loss. As has been proposed previously, the earlier rising activity in PM regions could reflect involvement in distinct aspects of movement, ones that require more pre-movement preparation (*Graziano, 2009*).

## Manifestations of hierarchy in firing patterns

Our results indicate that asymmetries in the reciprocal influence between regions may not manifest at the level of firing patterns in ways we might have expected. In particular, we found a substantial imbalance in reciprocal influence but also that firing in each region was dominated by patterns shared between regions and was equally predictive of future firing in the other region at the single-neuron level. These results raise questions about contemporary attempts to infer interactions between regions through network modeling of activity patterns alone. To serve as inputs that induce observed activity patterns, network models may exploit similar activity patterns seen in different regions. In a measurement regime where a very small fraction of relevant neurons are sampled, models could attribute influence that is actually local, but from unsampled neurons, to neurons with similar activity patterns observed in other regions. Models may then fail to reflect functional relationships between neurons that are revealed by activity perturbations. One way to at least partially address this may be to enforce in models a balance of across-region and within-region inputs that agrees with observations.

Using DLAG, we detected that some observed activity can be captured by similar but delayed patterns with either region's activity leading, and RFA's activity led to a greater extent. This generally agrees with the notion of a partial motor cortical hierarchy, with greater influence from the premotor RFA to the primary motor CFA than vice versa. It may then be tempting to interpret these activity components as mediating the asymmetric reciprocal influence we observed from optogenetic perturbations, but this remains supposition. We should also consider why other analyses we performed did not detect asymmetries like those we observed with DLAG. The failure to detect asymmetry with CCA

and PLS applied after shifting one region's activity relative to the other's may indicate that the across-region components detected by DLAG, though differing in activity variance capture, can still accommodate similar correlation or covariance when activity is aligned at a lag. In addition, the firing pattern predictivity calculations we performed for pairs of neurons may belie features of firing that emerge at the population level and are detected by methods like DLAG. Importantly, the predictivity measures we used here do not account for the fraction of population-level activity variance each neuron's firing pattern reflects. Consider a neuron whose time-varying firing rate is twice that of another: its firing pattern could be roughly equally predictive of any other pattern (any effect of the increased number of spikes could be relatively small), but it would reflect twice as much population-level activity variance.

It is commonly assumed that feedforward influence between regions will create predictive relationships between the activity patterns in the upstream and downstream regions. However, we have found that this does not hold on the scale of neuronal populations in the motor cortex during reaching in mice. This helps explain previous findings from firing pattern analysis that have seemed at odds with notions of even a partial motor cortical hierarchy. For example, analyses of activity in PM and M1 have described highly similar firing patterns (*Kimura et al., 2017*; *Riehle and Requin, 1993*; *Hyland, 1998*), similarities in preparatory activity (*Elsayed et al., 2016*), and symmetry of modeled reciprocal input (*Truccolo et al., 2010*). Our results show that these observations do not contradict an actual imbalance in reciprocal functional influence between regions. In general, our results from firing pattern analysis suggest that care should be taken when interpreting the results from any single analysis in relation to system function. The accurate inference of interactions between populations may require accounting for more than just simultaneous activity patterns.

We note here that our firing pattern analysis has involved all recorded activity; it is possible that results may differ if we focused specifically on a subset of activity components that correlate strongly with muscle activity. However, if the symmetry we observed for firing pattern predictivity or CCA/PLS was replaced by an asymmetry for this subset of activity components, our results would imply an opposite asymmetry for the remaining activity components, which may be surprising.

## Models of PM-M1 interaction

From many different random initial conditions, our dual-population network fitting converged on solutions that well-fit mean firing patterns across our three dataset types – recording from either region while inactivating the other, and paired recording of both regions without inactivation – and exhibited a difference in the dependence of activity in each region on across-region input. Here, we were not able to directly fit the observed activity of individual neurons, since our three dataset types were collected in different mice. However, our results provide a proof of principle that population-level features we observed, like asymmetric reciprocal influence and earlier activity change in one population, can emerge from local population dynamics, rather than asymmetric across-region input. This suggests that suppositions about how functional influence between populations may be reflected in firing patterns may not survive empirical testing, as the dominance of local population dynamics can decouple firing patterns from the functional influence between populations. These results also suggest caution in interpreting the relative strength of inputs between populations in network models, whether input weights or weights times activity, in relation to system function.

Our findings comport with recent theoretical results suggesting that local circuit dynamics can be the primary determinant of firing patterns in the presence of substantial across-region input (*Bachschmid-Romano et al., 2023*; *Gozel and Doiron, 2023*). Across-region connections may instead function to coordinate firing patterns across regions or modulate higher-order firing pattern features, like their autocorrelation (*Chaudhuri et al., 2015*). Firing patterns may be defined primarily by other constraints, such as the need to generate appropriate motor output patterns. The propagation of local activity in the service of this systemic pattern generation could be another constraint, as a number of recent observations suggest that local circuit dynamics govern motor cortical firing (*Shenoy et al., 2013*; *Seely et al., 2016*) to some extent (*Sauerbrei et al., 2020*). Our results underscore the recognized need (*Gozel and Doiron, 2023*) for developing theory about across-region interactions.

## Methods

### Experimental animals

All experiments and procedures were performed according to NIH guidelines and approved by the Institutional Animal Care and Use Committee of Northwestern University. A total of 26 adult male mice were used, including those in early experimental stages to establish methodology. Strain details and number of animals in each group are as follows: 21 VGAT-ChR2-EYFP line 8 mice (B6.Cg-Tg(Slc32a1-COP4*H134R/EYFP) 8Gfng/J; Jackson Laboratories stock #014548); and 5 C57BL/6 J mice (Jackson Laboratories stock #000664). All mice used in experiments were individually housed under a 12 hr light/dark cycle. At the time of the measurements reported, animals were 17–22 weeks old. Animals weighed approximately 23–28 g. All animals were being used in scientific experiments for the first time. This includes no previous exposure to pharmacological substances or altered diets.

### Directional reaching task

We modified a recently published directional reaching task (*Galiñanes et al., 2018*). Head-fixed male mice were trained to reach to one of four spouts to grab a water reward they could then bring to their mouth and ingest. Mice initiate trials by placing their right hand on a rung (during training) or relaxing their forelimb muscles (during neural recording) for a period randomly chosen for each trial between one and three seconds (rest period). One of four LED lights in front of the mouse illuminates at trial onset, the location of which corresponds to the randomly-selected spout location where water will be dispensed on the given trial (*Figure 1A*). If the forelimb remains at rest for the duration of the rest period, a 4 kHz tone ('Go' cue) sounds for 100ms, a water droplet is dispensed, and the mouse is free to reach out and grasp it. If the mice move their forelimb before the rest period ends, a 400 Hz buzzer sounds for 50ms, the LED turns off, and a 200ms delay must pass before another trial can be initiated. If the water droplet is not retrieved within 1 s (response period), it is removed by a suction tube immediately beneath the dispensation spout and the buzzer sounds. If mice reach the correct spout, a 2 s consumption period is imposed before a subsequent trial can be initiated. If the mice reach the incorrect spout first, the buzzer sounds for 50ms.

### Apparatus

The training apparatus was housed inside a sound-attenuating chamber (H10-24TA, Coulbourn). Head-fixed mice were positioned within a 3D printed enclosure with sections removed to allow fixation of the mouse's headplate to a headplate holder and to allow the right hand access to the rung and spouts. Enclosures also had a second rung for the left (non-reaching) hand and a divider below the mouse's chest that extended in front of the mouse to prevent the left hand from gaining access to the spouts. The waterspouts (blunted 21 G needles) were positioned in front of the mouse on its right side in a diamond configuration. The waterspouts were 6 mm apart vertically and horizontally. Suction tubes (blunted 21 G needles) that were 1.5 mm shorter were attached below each spout. The spouts were secured by a 3D printed holder, and their position was adjusted using a three-axis manual micromanipulator (UMM-3C, Narishige). The spouts and suction tubes were connected to solenoid valves (161T012, Neptune Research) through flexible tubing (EW-06422–01, Cole-Parmer).

Waterspout and rung touches were detected with capacitive touch sensors (AT42QT1011, SparkFun) during training. Capacitive touch sensors could not be used during neural recording since they cause artifacts in recorded voltages. Thus, during neural recording, spout touches were instead detected with infrared beam sensors positioned to detect a hand just in front of each spout (FT-KS40 and FX-502, Panasonic), and right forelimb muscle activity measured with EMG electrodes was used to initiate trials. Mice initiated trials by reducing muscle activity below a threshold set so that any limb movement during the rest period before the Go cue would cause a threshold crossing and abort the trial. The buzzer was also removed during neural recording.

Experimental control was performed using the Arduino Due. Four speakers and an Arduino Uno were used to play the reward tone. Four green LEDs (1.8 mm diameter) were secured by an LED holder in a diamond configuration at 10 mm apart vertically and 20 mm apart horizontally. The LEDs were positioned approximately 10 mm away from the front of the mouse's eyes and at the same height.

## Training

Under anesthesia induced with isoflurane (1–3%; Covetrus), mice were outfitted with 3D printed head plates (24x20 x 5mm) affixed to the skull using dental cement (Metabond, Parkell). Headplates had an open center that enabled subsequent access to the skull, which was covered with dental cement. During headplate implantation, the position of bregma relative to marks on either side of the head-plate was measured to facilitate the positioning of craniotomies during later surgeries, or the skull was covered with clear cement to maintain the visibility of bregma.

After recovery from headplate implantation surgery, mice were placed on a water schedule in which they received 1 mL of water per day. At least 4 days after the start of the water schedule, mice were acclimated to handling by the experimenter and head-fixation using a modification of established procedures (*Guo et al., 2014a*). After a day of acclimation to handling, mice were acclimated to head-fixation and the reaching task over 1–4 daily sessions during which they were head-fixed in the reaching apparatus and provided water rewards.

During head-fixed acclimation, the water droplets were dispensed from all 4 spouts at random intervals (5.5–6 s). The four spouts were placed in front of the mouth, and all spouts dispensed a 5 µL water droplet signaled with the reward tone. The four LEDs all turned on while the water was being presented. The water droplets were automatically removed by suction at the end of the interval regardless of whether the mouse collected them or not. Mice freely licked the waterspouts and quickly learned the association between sound and reward. Once this was learned, the waterspouts were moved to a lower right position that allowed easy access for the right hand. Mice spontaneously performed reach-to-grasp movements to collect and consume the water droplets by licking their hand. Almost all mice completely switched to the reaching behavior during the first or second session.

Following acclimation, mice underwent a daily 60 min training session to learn to initiate trials by touching the rung and to associate the location of the illuminated LED with the waterspout location where reward would be dispensed. To encourage mice to reach to all four spout locations, the probability of each spout being selected for water dispensation on a given trial was computed based on the percentage of successful reaches to the given spout for past trials during the given session. Probabilities were inversely proportional to success rates. In addition, two task parameters were adaptively changed during training sessions to shape mouse performance: the rest period was gradually lengthened (starting from 0.1 s), and the response period during which the water was available was shortened (starting from 10 s). Over 7–18 daily training sessions, mice learned to associate the illuminated LED location with the spout location, to wait during increasingly long rest periods, and to reach within increasingly short response periods. The rest period gradually became longer until mice were able to wait for more than 2 s. The response period adaptively changed until they were able to reach within 1 s.

For some mice, upon reaching these behavioral thresholds, neural recordings were performed during subsequent training sessions (n=3 mice, total of 10 recording sessions). For another cohort of mice, once they met these conditions, recordings were performed during subsequent sessions using an experimental control script that did not adaptively update task parameters. In this script, the probability of each spout being selected for water dispensation did not vary, the rest period randomly changed between 1 and 3 s in 0.2 s increments, and the response period was fixed at 1 s (n=3 mice, total of 11 recording sessions). These data were analyzed separately but eventually combined because there was no substantive difference in the results.

In order to be considered 'trained' and used for experiments involving optogenetic inactivation and/or Neuropixel recording, mice had to demonstrate the capacity to reach successfully to multiple spouts. Successful reaching to a given spout was defined as the mouse reaching to the spout much more often than expected by chance for trials rewarding the spout if a reach was attempted (trials where no reach was attempted were ignored). Chance was defined as 25% of trials with an attempted reach, since there were four spouts. The typical mouse reached this criteria for three of the four spouts.

## EMG recording

EMG electrodes were fabricated for forelimb muscle recording using established procedures (*Miri et al., 2017*; *Akay et al., 2006*). Briefly, each set consisted of four pairs of electrodes, each consisting of two 0.001" braided steel wires (793200, A-M Systems) knotted together. On one wire of each pair, insulation was removed from 1 to 1.5 mm away from the knot; on the other, insulation was removed

from 2 to 2.5 mm away from the knot. The ends of the wires on the opposite side of the knot were soldered to an 8-pin miniature connector (33AC2364, Newark). Different lengths of wire were left between the knot and the connector depending on the muscle a given pair of electrodes would be implanted within: 3.5 cm for upper forelimb muscles and 4.5 cm for lower forelimb muscles. The ends of wires with bared regions had their tips stripped of insulation and then were twisted together and crimped inside of a 27-gauge needle that facilitated insertion into muscle.

Mice were chronically implanted with EMG electrodes during the surgery in which headplates were attached as described previously (*Miri et al., 2017*; *Warriner et al., 2022*). Insertions targeted the biceps (elbow flexor), triceps (elbow extensor), extensor carpi radialis (wrist extensor), and palmaris longus (wrist flexor). As discussed previously, while our methods produce isolated recordings from antagonist muscle pairs, we cannot exclude the possibility that EMG recordings are influenced by the activity of nearby synergist muscles, since our methods do not readily allow for simultaneous recordings from synergist muscles in the mouse forelimb.

Recordings were amplified and bandpass filtered (1–75,000 Hz) using a differential amplifier (C3313, Intan Technologies). Data was digitized and acquired at 30 kHz using the RHD2000 USB interface board and RHD USB interface GUI software (Intan Technologies). Suprathreshold activity of any muscle was detected in this software to indicate forelimb movement. Typical thresholds were 150–500 µV.

## Optogenetic inactivation

After VGAT-ChR2-EYFP mice reached proficiency after several days of training on the directional reaching task, dental cement above the skull was removed and a 2–2.5 mm diameter craniotomy was made above the left CFA or RFA. A thin layer of Kwik-Sil (WPI) was applied over the dura, and a 4 mm diameter #1 thickness cover glass (64–0724, Warner Instruments) was placed on the Kwik-Sil before it cured. The gap between the skull and the cover glass was then sealed with dental cement around the circumference of the glass. A small 0.5 mm diameter craniotomy was then opened over the other region for recording.

During subsequent behavioral sessions, a 400 µm core, 0.39 NA optical patch cable (FT400EMT, Thorlabs) terminating in a 2.5 mm ceramic ferrule was attached to a micromanipulator (SM-25A, Narishige). We set the cable at a certain distance above the surface of the brain for each session using a micromanipulator to ensure that the cone of light emanating from the cable would project a spot of light 1.5 mm in diameter onto the surface of the brain. A Neuropixel probe was inserted into the open craniotomy, and recordings were performed as described in the next section, except the agarose and paraffin were omitted to avoid covering the window above the other region. To attenuate firing throughout motor cortical layers, we used a 450 nm laser (MDL-III-450–200 mW, Opto Engine LLC) to apply 50ms pulses of light at an intensity of 9 mW/mm2 to the brain surface. To inactivate the motor cortex near the outset of reaching, the light pulse was triggered when either the biceps or triceps EMG signal reached a threshold set to reflect activation above baseline inactivity (usually 100–500 µV). Light was applied during a random 50% of the trials on which the stimulation conditions were met. Unstimulated trials were used as controls.

## Neural recording

Acute recordings using two Neuropixel 1.0 probes (Imec) were completed as mice performed the reaching task. To expose the recording area, dental cement above the skull was removed and a 2 mm diameter craniotomy was made over CFA and a 1 mm diameter was made over RFA. The exposed brain tissue was sealed with silicone elastomer (DentSilicone-V, Shofu, or Kwik-Cast, World Precision Instruments). A stainless-steel screw (U-1415–01, Hirosugi-Keiki) was implanted above the contralateral cortex as a ground. The large tip electrode on the shank of Neuropixels was used as a reference. Before recording, the animal was head-fixed, the silicone elastomer was removed, and the Neuropixels were slowly inserted at an inclination of 30° in the coronal plane and 15° in the parasagittal plane using fine micromanipulators (SM-25A, Narishige). For CFA, probes were inserted between 1–2 mm lateral and 0.5 rostral to 0.5 mm caudal of bregma. For RFA, probes were inserted 0.35–1.35 mm lateral and 1.75–2.75 mm rostral of bregma. Once the probes were in place, the brain surface was covered by agarose gel (2% agarose-HGT, Nacalai Tesque) and a mixture of liquid and solid paraffin, to minimize

the vibration of the brain. Data was acquired at 30 kHz using the PXIe Acquisition Module (Imec) and SpikeGLX software (Janelia Research Campus).

## Quantification and statistical analysis
All analysis was completed in MATLAB versions R2021b or later (MathWorks).

## EMG processing and analysis
With certain exceptions discussed below, EMG measurements were downsampled to 1 kHz, high-pass filtered at 250 Hz, rectified, and convolved with a Gaussian having a 10ms standard deviation.

## EMG during optogenetic inactivation
Before EMG trial averages were analyzed, outliers were removed. The total Euclidean distance between muscle time series segments from −50 ms to 0 ms before light/trial onset, summed across muscles, was computed (MATLAB function 'pdist2'). Control and inactivation trials were combined for each muscle. The distances were then averaged across trials. A threshold was set one standard deviation above the mean. Trials above this threshold were excluded and were not used in subsequent EMG analyses.

The absolute difference between control and inactivation trials was calculated using the difference between the means for EMG time series for each muscle. The fractional changes in the time series were corrected for the difference expected by chance due to the use of separate sets of trials, estimated as follows. On 1000 different iterations, we divided the control trials into random halves and similarly calculated an absolute difference time series using the two halves. The mean absolute difference time series across these 1000 iterations was computed, and this mean was subtracted from the absolute difference time series computed with the actual data. Lastly, we subtracted the mean value across the 20ms preceding light/trial onset time from the resulting time series.

## Identification of reach initiation using EMG
Muscle activity measured from EMG recordings was used to determine the time of reach onset. For this analysis, EMG across all recorded muscles was summed to obtain a single trace representing overall muscle activity. Reach epochs were first identified using the reward tone and port beam break sensor traces to find trials where the animal successfully obtained the water droplet. As the animal's reaching strategy could trigger an additional beam break at an incorrect port, successful reaches were defined as instances when the beam break at the correct port occurred within 50ms of the first beam break at any port. As the rest period prior to the reward tone is at least 1 s in duration, the EMG baseline for each reaching trial was found by computing the average summed EMG between −500ms and −100ms relative to the reward tone. A global standard deviation from baseline during quiescence (no muscle activation) was identified by calculating the standard deviation of an arbitrarily selected quiescent period identified in the recording. A lower threshold for each trial is set to the EMG baseline plus 7 times the global standard deviation. The upper threshold is set to the 30th percentile EMG value across epochs from reward tone to beam break for all successful trials. The reach onset time was defined as the lower threshold crossing immediately preceding the first upper threshold crossing after the tone. The double threshold algorithm was used because on some trials, EMG would first go above and then return below the lower threshold before the reach, likely from a twitch or other involuntary reaction to the reward tone. Trials in which the EMG never crossed the upper threshold were excluded from further analysis.

## Reaching trial exclusion
Due to the presence of successful reaching trials with outlying muscle activity traces, certain exclusionary criteria were implemented to curate a collection of reaching trials to be utilized for subsequent analysis. Reach duration was calculated by subtracting the reach onset time from the beam break time. Successful trials in which the reach duration was longer than the mean duration of all successful trials in the given session plus 3 times the standard deviation were flagged for exclusion. Reaction time was calculated by subtracting the reward time (solenoid command pulse onset) from the reach onset time. Successful trials in which the reach onset was less than −50ms were flagged. Negative reaction times were possible in sessions where a fixed rest period was used because the animal was

able to predict the reward tone. Ramp time was calculated by subtracting the reach onset time from the second threshold crossing time. Successful trials in which the ramp time was longer than the mean ramp time for all successful trials in that session plus three times the standard deviation were flagged. Finally, successful trials in which the standard deviation of the trial's individual EMG baseline was greater than the mean baseline standard deviation for all successful trials in that session plus three times the standard deviation were flagged. The final set of qualifying successful trials for each session was obtained by removing any flagged trials.

## Spike sorting and unit curation

Putative spikes were detected and sorted with Kilosort3 (*Pachitariu et al., 2016*) using default parameters. Sorting was improved (i.e. single unit yield was increased) by excluding pathological channels before sorting. These channels were identified using abnormalities in high-frequency components. The fast Fourier transform (MATLAB function 'fft') was computed for each channel, and the weights over the interval from 300 to 1000 Hz were summed. We observed that the magnitude of this sum varied smoothly across channels, with the exception of certain channels which were characterized by much larger, outlier values. After median subtracting over a window of 10 channels, pathological channels were identified using a threshold of the median plus or minus three times the standard deviation. Channels identified in this manner for exclusion were consistent across recordings for the same probe. We excluded between 5 and 10 channels per session.

Single units identified by Kilosort3 were further curated by removing those with abnormal refractory violations. Inter-spike intervals (ISIs) were calculated by taking the autocorrelation of the spike trains binned at 0.1ms. A test statistic was generated by summing ISI frequencies from 0.3 to 1.0ms and normalizing by the summed ISI frequencies between 10 and 50ms to account for overall firing rate. Units with a test statistic greater than 0.18 were excluded from any further analysis. This threshold is based on the frequency of violations expected in a sorted unit resulting from two neurons that have Poisson spiking and with one of the units accounting for 90% of the spikes.

Depths of sorted units were analyzed to identify cortical cells. CFA has a thickness of approximately ~1500 μm, and RFA has a thickness of ~1800 μm. For both CFA and RFA recordings, the depth of the most superior unit was used as an approximation for the cortical surface. Units within 1500 μm of this superior unit were considered cortical cells and subsequently analyzed. Firing rates were estimated at each ms during recordings by summing Gaussians with a 10ms standard deviation centered on each spike time.

Putative pyramidal cells were identified based on waveform width. Widths for each unit were calculated by finding the trough-to-peak duration of the assigned waveform template. A histogram of waveform widths collected across all dual-probe recording sessions exhibited a bimodal distribution that could be well approximated by the sum of two Gaussians. Using this model, we defined a width threshold that permits a 5% misclassification rate where tails of the fitted Gaussians fall above or below the threshold (*Miri et al., 2017*). A threshold of 0.417ms was defined as the difference between wide-waveform and narrow-waveform cells.

Units were also classified based on whether their peak or trough had a greater magnitude. This was determined by analyzing the assigned waveform template and determining whether the maximum absolute value was originally positive or negative.

## Identification of neurons significantly correlated with muscles

In order to identify the neurons that significantly correlated with muscle activation, instantaneous firing rates were calculated from spike trains by smoothing each train with a 10ms standard deviation Gaussian. This allowed the correlation to be measured between a neuron's instantaneous firing rate time series with each muscle's recorded EMG activity. A bootstrapping approach was implemented to determine which of these correlations were significant and not by chance. Spike trains were circularly shifted at least 10 s and no more than 10 s less than the duration of the experiment (to ensure that the circular shift adequately removed any temporal structure between the neurons and the muscles), and correlations were calculated again. This process was repeated 300 times producing a null distribution for each neuron's correlation 'by chance' to each muscle. The observed correlations for each muscle were measured against the null distributions using a two-tailed test (to find both correlated

and anti-correlated neurons), and if any p value corresponding to a muscle was significant (p<0.05), the neuron was labeled as significantly correlated to muscle activity.

## Optogenetic inactivation effects on firing patterns

For analysis of optogenetic inactivation effects, outlier trial exclusion was done differently since the activity perturbation may affect trial outcome. This exclusion was done by combining both inactivation and control trials for each neuron. We determined the distance between firing rates over the 20ms before laser/trial onset between all the trials (MATLAB function 'pdist2') under the assumption that before reach initiation, the firing rate series for each neuron should look similar. These distances were then averaged across neurons for each trial. A threshold was set at the mean of these distances plus a quarter of the standard deviation. Trials falling above this threshold were excluded.

Trial-averaged neuronal firing rates for control and inactivation trials were baseline subtracted by subtracting the mean firing rate from –20–0ms before light/trial onset from the entire time series for each neuron. Across-animal trial averages were calculated using the same number of neurons from each animal to prevent animals with more recorded neurons from dominating the averages. We first found which mouse of the three similarly inactivated had the lowest number of recorded neurons, n, and used n randomly chosen neurons from the other two mice.

The absolute difference between inactivated and control trials was calculated for individual neurons and averaged across similarly inactivated animals for comparison. We subtracted the control trial average from the inactivation trial average for each neuron, and then took the absolute value. The averages again used the same number of neurons from each of the three mice based on which-ever had the least amount recorded. The resulting average difference time series for RFA- and CFA-inactivated mice were baseline corrected by subtracting the mean change from –20–0ms before laser/trial onset from the entire time series.

To verify that Channelrhodopsin2 (ChR2) stimulation of inhibitory interneurons only occurred in the target region and did not spread to the recorded region, we modified SALT (*Kvitsiani et al., 2013*), a method developed to detect light-responsive neurons in a statistically based, unsupervised manner. A window size of 5ms was used for baseline and test epochs, and the analysis used a time resolution of 1ms. In order to ensure that the resulting p-values followed a uniform distribution from 0 to 1 under the null hypothesis of no direct light effect, a random sample from the Jansen-Shannon divergence values obtained from the post-stimulus firing patterns was used for comparison to an empirical null distribution based on firing during non-stimulus epochs. This step replaced the step in the original algorithm where the median of the Jansen-Shannon divergence values was taken, which did not yield a uniform p-value distribution under the null hypothesis. For each recording session, a p-value was computed for the firing of each narrow-waveform neuron following light onset, where a low p-value indicates a neuron more likely to directly respond to light. For our purposes, the approximately uniform distributions of p-values we observed (*Figure 3—figure supplement 1C–F*) demonstrate general conformance to the null hypothesis of no direct ChR2 activation in the recorded region.

## Delayed latents across groups (DLAG)

The application of DLAG used firing patterns during qualifying successful trials (see above). To avoid covariance matrix rank deficiency in fitting DLAG models, we excluded low-firing neurons by setting a threshold such that any neuron that spiked fewer times than half the number of trials was excluded. For remaining neurons simultaneously recorded in a given session, firing rate matrices were assembled for each region, where each element was a given neuron's firing rate average over a 20ms window. Trials spanned from 200ms before to 800ms after reach initiation. Matrices were thus N neurons x 50 time bins. The DLAG model was then fit using these matrices for each trial. Positive delay values indicated latent variables where CFA led RFA, and a negative delay value indicated latent variables where RFA led CFA.

To probe the robustness of observed results, we varied the numbers of across-region and within-region latent variable dimensionalities (i.e. the numbers of latent variables). Models were fit to each session's data with every combination of across-region dimensionalities of 2–6 and within-region dimensionalities of 2–6 in each area; therefore, sessions were eligible only if each area had at least 12 remaining neurons (n=15 sessions). Any across-region latent variable with a delay that failed to

converge or reached the boundary values of ±200ms in 10,000 iterations was removed from the analysis.

To test if identified delays were significantly different from zero, bootstrapping was performed by sampling 100 trials from the set of existing trials with replacement, as in the originally published method (*Gokcen et al., 2022*). A delay was considered statistically significant if less than 5% of the bootstrapped samples performed just as well with a model with a 0ms delay as they did with the model with the original calculated delay. Across-region latent variables with non-significant delays were removed from the analysis. For each session, we calculated the proportion of variance captured by across-region latents with statistically significant delays, grouping latents by the sign of their delays (i.e. which region led).

## Canonical correlation analysis (CCA) and partial least squares (PLS)

CCA (MATLAB function 'canoncorr') and PLS were performed on matrices comprising the trial-averaged firing rates for all neurons from an animal, concatenating averages aligned on both reach onset and spout contact for reaches to each of the four spouts (i.e. eight trial averages are concatenated). Averages spanned from 99ms before to 100ms after the alignment point, resulting in matrices of size 1600 time points x NCFA neurons and 1600 time points x NRFA neurons for each recording session, where NCFA is the number of included neurons from CFA and NRFA is the number of included neurons from RFA. We denote these matrices XCFA and XRFA in what follows. PCA was performed on these matrices, and on average, 25 principal components were able to capture 95% of the neural activity variance, and so 25 was chosen as the number of PLS components and canonical variables for the subsequent analyses, enabling comparison to PCA results.

To mitigate matrix rank deficiency for CCA, principal component analysis was performed on the neural activity matrices and the first 25 principal components were then used for the CCA. Since the resulting 25 canonical vectors (CVs) for each data matrix are not necessarily orthogonal, they were orthogonalized in order to compute the additional neural activity captured by successive CVs, enabling comparisons with principal components. Weighted averages of Pearson correlations for all pairs of CVs were computed by weighting the Pearson correlation of each CV pair by the average of the additional neural activity variance captured by each corresponding orthogonalized CV from the pair. To account for synaptic delay between RFA and CFA, lags were introduced in the same way as stated below for the PLS analysis.

For PLS, the PLS-SVD variant *Le Floch et al., 2012* was used. PLS-SVD is done by performing singular value decomposition on the cross-covariance matrix $X^TY$, which in our case is $X_{RFA}^T X_{CFA}$, yielding.

$$USV^T = X_{RFA}^T \cdot X_{CFA}$$

where the columns of $U$ define axes in RFA activity space and the columns of $V$ define axes in CFA activity space. The matrix $S$ is a diagonal matrix where the $n^{th}$ diagonal element is the covariance of RFA activity projected onto the $n^{th}$ column of $U$ and CFA activity projected onto the $n^{th}$ column of $V$. The trace of $S$ can be interpreted as how much total covariation between RFA and CFA activity is captured by the PLS components. A necessary additional step is to divide this number by the total variance in neural activity in these two matrices in order to normalize by how active these brain regions are during the reach and grasp epochs that are being analyzed.

$$c = \frac{\text{trace}\,(S)}{v_{RFA} + v_{CFA}}$$

where $v_{RFA}$ and $v_{CFA}$ are the sum of the variances of the rows of $X_{RFA}$ and $X_{CFA}$ respectively. The original activity matrices can then be projected onto their respective axes (by $X_{RFA} * U$ for RFA, $X_{CFA} * V$ for CFA) to produce time courses in neural activity space such that corresponding pairs will maximally covary, that is the projection of $X_{RFA}$ onto the $k^{th}$ column of $U$ and $X_{CFA}$ onto the $k^{th}$ column of $V$ will have covariance equal to the $k^{th}$ diagonal element of $S$.

PLS-SVD was similarly performed after shifting one activity matrix in time relative to the other. This was done by keeping the same $X_{RFA}$ matrix for each animal, but constructing a new $X_{CFA}$ matrix. The reach and grasp onset times were shifted by an amount between –30 and 30ms and the trial epochs –99–100ms around each shifted time point were extracted, trial averaged, and concatenated just as

before. PLS-SVD was then performed, and $c$ was recomputed for each lag from –30 to 30 for each animal.

An additional control analysis was performed to test whether the similarity between CFA and RFA activity found with CCA and PLS was beyond what could be expected by chance for activity measured and processed according to our approach. To do this, we repeated either alignment method, keeping $X_{CFA}$ the same, but shifting each reach and grasp onset used for aligning trials for $X_{RFA}$ by a random number of milliseconds between 5000 and 10000 into the future. Checks were implemented to make sure that each shifted trial did not overlap with another (original or new) trial. This removes the behavior-related activity correlations between the original matrices used for alignment.

## Firing pattern predictivity measurements

These measurements were performed on a subset of simultaneously recorded neuron pairs from dual probe recordings. Only wide-waveform, putative pyramidal neurons having a maximum waveform deflection in the negative direction were used. Average firing rate was calculated for each neuron across all qualifying successful trials over the window from 200ms before to 800 ms after reach initiation. The ten thousand neuron pairs with the highest average firing rate products across all sessions were identified. Note that pairs only included neurons that belonged to the same session, as simultaneous recording was required for the trial-by-trial predictivity analysis performed here.

After calculating the firing rate distribution of the CFA and RFA cells belonging to the top ten thousand pairs, CFA cells had higher firing than RFA cells on average. Firing pattern prediction can be heavily influenced by differences in firing rate, since statistical power can be related to the number of spikes observed. Thus, we developed an algorithm to match the distributions of firing rates by excluding certain pairs. We first generated histograms of the log firing rate for the CFA and RFA cells belonging to the top ten thousand pairs binned at 0.05 log spikes per second. Note that the multiplicity of a given neuron in these distributions matched the number of pairs the given neuron appeared in. To start the algorithm, a firing rate bin with more CFA cells than RFA cells was chosen. We then randomly selected a cell pair that contained a CFA cell from this bin and an RFA cell from a firing rate bin which had more RFA cells than CFA cells. To better match the firing rate distributions, the high firing rate CFA cell was then replaced with a different CFA cell from the same session and firing rate bin as the selected RFA cell. The selection of new CFA cells was allowed to occur with replacement. This selection process was repeated until all firing rate bins contained the same number of CFA and RFA cells, or when no other cell pairs could be created to match the binned firing rate distributions.

From distributions of p-values calculated as described below for each metric, we estimated the fraction of false null hypotheses as one minus an estimate of the a prior*i* fraction of true null hypotheses (*Storey, 2002*) (MATLAB function 'mafdr').

## Transfer entropy

Transfer entropy analysis was performed using a publicly available algorithm (*Ito et al., 2011*). This algorithm takes the spike trains of two neurons (one source neuron, one target neuron) and compares how well the spiking of the target neuron can be predicted when considering its past history as well as the past history of the source neuron. The better the source neuron's past history improves the prediction of the target neuron's spiking activity compared to the source neuron's own past history alone, the higher the transfer entropy (TE) value. In this case, to increase our statistical power, rather than using the trial segments used for Granger causality described below, we extracted and used spiking during epochs of movement in the following way. First, rectified and filtered EMG recordings from the four muscles were summed. Then, a period of approximately one second of muscle quiescence was manually identified in this summed time series. A threshold was set at the average value over this period plus 7 times the standard deviation over this period, and movement epochs were defined as when summed muscle activity surpassed this threshold. To include brief reductions in muscle activity that happen during ongoing movement, any epoch of 100ms or less below this threshold was reclassified as movement. Additionally, periods of movement less than 10ms were reclassified as non-movement since this was unlikely to reflect meaningful movement. The neural activity during these movement epochs was then extracted and concatenated. However, when concatenating epochs, the discontinuous skips in time from the end of one epoch to the beginning of the next could, although very sparse, cause inaccurate calculations of the transfer entropy by counting spikes from one time point

as causing spikes at a time point much farther in the future. To circumvent this issue, pads of zeros were inserted between each movement epoch so that no spikes in one epoch could be calculated as causing spikes in the next movement epoch. To calculate transfer entropy, we used TE$_{J->I}$(d) from eq. 5 in *Gokcen et al., 2022*. We calculated TE$_{J->I}$(d) for d=0,1,2,...,30 ms and took the maximum value as our measurement for the transfer entropy between two neurons, the observed TE value.

In order to determine the significance of the observed TE value for each neuron pair, we obtain a corresponding p value by creating empirical null distributions, that is TE values expected in the absence of coupled firing between the neurons. For each neuron pair, the concatenated spike train of the source neuron is circularly permuted (MATLAB function 'circshift') by values no less than 3 s and no more than T-3 seconds (where T is the duration of the concatenated spike train) so that any temporal relationship between the two spike trains is extinguished while preserving the spiking statistics and firing rate patterns of each individual spike train. After this permutation, the TE was measured as before. This process was done 300 times to form a null distribution of TE values for each neuron pair, and a p value was calculated by taking the fraction of null TE values that were higher than the observed TE value.

## Granger causality

Point-process Granger causality with exogenous temporal modulations (GC *Casile et al., 2021*) was utilized to assess directional bias in the predictivity of neuronal firing between CFA and RFA. Using GC, predictive models are generated to fit spiking data of a target neuron using the spiking data from the entire neuronal ensemble, both with and without the source neuron. In our conditions, this neural ensemble contains neurons from both CFA and RFA. The method yields a test statistic for each source-target neuron pair by computing the difference in accuracy of the prediction from these two models. The larger the difference, the more unique information the source neuron contains about the spiking of the target neuron. This test statistic follows a chi-squared distribution, yielding p-values for each source-target pair.

Binary spike trains for each neuron belonging to the top ten thousand firing rate CFA/RFA neuronal pairs were generated. Only spiking data between 200ms prior to reach initiation and 800ms after reach initiation were used. GC was conducted on a per-session basis using a random selection of 40 qualifying successful trials (the same for all pairs) to limit computation time. Global regression of exogenous temporal modulations was optimized to bins of either 1ms, 5ms, 8ms, 10ms, 20ms, or 25ms. The duration of spiking history incorporated in the model was fixed at 30ms to force all computed test statistics to follow the same chi-square distribution.

For certain source-target pairs, the model did not converge on the target spiking data within tolerance. We removed these pairs from the analysis. Additionally, for some pairs, the model would be overfit and predict the target data perfectly. We also removed these pairs from further analysis. As a consequence, 35% of the 10,000 pairs with the highest product of average firing rates were excluded from the GC analysis for these reasons. However, this should not affect our results in a meaningful way. While it could affect the degree of skew toward 0 in p-value distributions, here we are not ascribing significance to the degree of skew, just the similarity between the skew for CFA source and RFA source conditions. We also do not draw distinctions between the results from different prediction methods that would be affected by only using a subset of neurons for GC.

Following these exclusions, we observed a preponderance of pairs with p-values greater than 0.95. We found that the firing rate product of these high p-value pairs was heavily skewed towards low values (*Figure 6—figure supplement 2A*). These low firing rate pairs likely did not meet the assumptions of GC and were removed from further analysis. Following this exclusion, the firing rate distributions of RFA and CFA neurons belonging to this final set of neuronal pairs were reanalyzed and found not to be appreciably different (*Figure 6—figure supplement 2B and C*).

The GC algorithm additionally computes the model difference test statistic for all within-region pairs. Using these values, the predictive power of the CFA and RFA intraregional neuronal population was also assessed.

## Convergent cross mapping

CCM was implemented using a modified version of MATLAB function 'Sugi' (*Krakovská et al., 2015*). Here, we used the algorithm to quantify how well the activity of the source (causal) neuron can be

predicted by the historical dynamics of a target (effector) neuron. While TE and GC assume the observations underlie a purely stochastic system, CCM can uncover causal interactions in a dynamical system with a weak to moderate deterministic component and where causal variables do not contain unique information.

For each pair of source and target neurons, the historical dynamics of the target neuron can be represented by a 'shadow manifold', where every point on the shadow manifold is constructed from time-lagged observations. Intuitively, we can think of nearby points on the shadow manifold as having similar dynamics. Thus, if the source neuron has causal interactions with the target neuron, then the activity of the source neuron at time t can be predicted from a cluster of nearby points at time t in the shadow manifold. CCM quantifies these causal interactions by finding the correlation coefficient between the predicted source activity calculated using a cluster of nearby points in the target manifold at time t and the actual source activity at the same time, a metric termed 'cross-map skill' (*Sugihara et al., 2012*; *Ye et al., 2015*).

The preprocessing and trial extraction followed the same steps used for Granger causality calculations. Five thousand RFA-CFA neuron pairs were curated by taking the top ten thousand pairs described above and excluding the bottom 50% when sorted by lower firing rate of the pair. This avoided anomalous results from CCM calculations that seemed to depend on epochs with very few or no spikes. Spike trains were then binned at 1ms and smoothed using a Gaussian kernel with a sigma of 10ms. For every neuron pair, 30 trials were randomly sampled from all qualifying successful trials to reduce computation time, and shadow manifolds were constructed from the previous 500ms of activity in the target neuron within the same trial. CCM was then applied to these 30 trials to calculate a cross map skill value, quantifying the correlation between the time series of predicted source activity and the time series of actual source activity. To account for non-instantaneous interactions between neurons, this process was repeated at time delays up to 30ms, incrementing every 3ms (i.e. 0, 3ms…, 30ms), keeping the maximal cross map skill value and its associated delay.

In order to determine whether these observed cross-map skill values were significant, an empirical null distribution was generated by disrupting the temporal relationship between pairs of neurons. For every pair, we repeated the same circular permutation method used with transfer entropy (300 permutations) and applied CCM to generate cross-map skill values, making sure to use the same 30 trials and optimal time delay used to obtain the actual cross-map skill for the given pair. P-values for each pair were obtained by counting the fraction of null cross map skill values that were greater than the observed cross map skill.

Applying CCM involved choosing several hyperparameters. E, the embedding dimension of the shadow manifold, can be thought of as the number of time lags that optimally captures the historical dynamics of the target neuron; if E is set to 4 then each dimension on the shadow manifold represents the neuron's activity at times [t, t-1 ms, t-2 ms, t-3 ms]. To determine the optimal value for E, we applied a simplex projection to each individual neuron in CFA and RFA to determine how many lagged dimensions best forecast a neuron's own future activity (*Clark et al., 2015*). This optimal embedding dimension for both CFA and RFA neurons was found to be 4. The number of nearby neighbors, K, searched for in the target shadow manifold was kept at 5 (E+1). 1D70F, which indicates how many timesteps each shadow manifold dimension is lagged in time [t, t-1*1D70F ms, t-2*1D70F ms…], was set to 1.

## Network modeling

A recurrent neural network was trained to replicate experimental data. The package PsychRNN *Ehrlich et al., 2021* was used to configure the network architecture, inputs, outputs, training, and simulation in the following way. Two neural populations of 500 units each, one representing RFA and the other representing CFA, were created with 80% excitatory units and 20% inhibitory units. The connection probability between neurons within each region was 5% and additional sparse connections were created from excitatory units in each region to neurons in the other region with a connection probability of 0.1%; no inhibitory units projected to the opposite region. A subset of 100 excitatory units from each region was chosen to be the units whose activity was trained to reproduce the observed muscle activity traces; these neurons were also stipulated not to have any projections to the opposite brain region. The simulated muscle activity output was the weighted sum of the activity of these 200 neurons, with weights allowed to vary during training. A second subset of 100 excitatory units from

each region was chosen to be the units whose activity was trained to reproduce the experimental neural activity traces.

Experimental data used for training all models consisted of averaged neural or muscle activity spanning from 50 before to 100ms after reach onset from successful reaches. Averages from one animal were used for each of three conditions (CFA inactivation, RFA inactivation, dual recording/no inactivation). However, since animals were used for only one type of experiment, training data reflected data from three different animals.

The simulated summed neural activity for each region was the weighted sum of each set of 100 neurons, with weights held constant to replicate how experimentally observed averages were calculated. To simulate activation of inhibitory interneurons expressing ChR2, all inhibitory units in the 'inactivated' region also received a 'light' signal as input. We modeled the light signal by concatenating two sigmoid curves, each with a maximum value of 1, using the following equation:

$$s(t) = \pm \frac{1}{1 + e^{-kt}}$$

The ± denotes that the sign of the numerator of the two sigmoids varies. The first sigmoid had a positive numerator and ramps up from 0 to 1, while the second had a negative numerator so that it starts at 1 and decays to 0. The first sigmoid had k set to 0.5 so that the curve started to ramp up (surpassing a value of 0.01) at reach onset and reached 0.95 15ms later. The signal sustained its maximum value of 1 until 50ms after reach onset, and the second sigmoid had k=1 so that the signal had a value <0.01 5ms later. All units also received a 'Go' signal as input in each of the three conditions. The 'Go' signal similarly was a sigmoid function that started at a value of 0 and had k=0.3, such that the curve started to ramp up (surpassing a value of 0.01) 40ms before reach onset, reached 0.95 25ms later, and then sustained its maximum value of 1 for the remainder of the trial.

Training consisted of up to 100,000 trials where each was randomly selected to be one of the three conditions. For each trial, the output of the muscle output units was measured along with the neural activity output, either from the non-inactivated brain region or both regions if the trial was a no-inactivation trial. The mean squared error between the model outputs and experimentally observed traces was computed and used as input to the cost function, which was minimized as model weights were updated. Weights for connections between units within and across regions were allowed to update, but certain constraints were applied. Positive weights remained positive and negative weights remained negative in order to keep the number of excitatory and inhibitory units constant. Weights between units that were initialized to be 0 remained 0 throughout training to keep the number of connections within and across regions constant. In the symmetric case, an additional term was added to the cost function: the absolute value of the difference between the sums of the weights of the across-region connections in each direction (RFA→CFA and CFA→RFA). As mentioned previously, output weights from the muscle output units were allowed to fluctuate throughout training but output weights from the neural activity output units remained fixed. Biases on all units also remained fixed during training.

## Acknowledgements

We are grateful to J Glaser and L Pinto for helpful conversations, and J Glaser and M Elbaz for comments on the manuscript. ASI was supported by the Japan Society for the Promotion of Science and the Uehara Memorial Foundation. AM was supported by a Searle Scholar Award, a Sloan Research Fellowship, a Whitehall Research Grant Award, The Chicago Biomedical Consortium with support from the Searle Funds at The Chicago Community Trust, the Simons Foundation, and NIH grant DP2 NS120847.

## Additional information

### Funding

| Funder | Grant reference number | Author |
| --- | --- | --- |
| National Institutes of Health | DP2NS120847 | Andrew Miri |
| Japan Society for the Promotion of Science | | Akiko Saiki-Ishikawa |
| Uehara Memorial Foundation | | Akiko Saiki-Ishikawa |
| Searle Scholar Award | SSP-2019-113 | Andrew Miri |
| Sloan Research Fellowship | FG-2020-13518 | Andrew Miri |
| Whitehall Research Grant Award | 2018-12-108 | Andrew Miri |
| The Chicago Biomedical Consortium with support from the Searle Funds at The Chicago Community Trust | C-098 | Andrew Miri |
| Simons Foundation | 875841 SCGB Pilot | Andrew Miri |

The funders had no role in study design, data collection and interpretation, or the decision to submit the work for publication

### Author contributions

Akiko Saiki-Ishikawa, Conceptualization, Investigation, Methodology, Writing – original draft; Mark Agrios, Sajishnu Savya, Adam Forrest, Diya Basrai, Investigation, Methodology, Writing – original draft, Writing – review and editing; Hannah Sroussi, Feihong Xu, Investigation, Methodology; Sarah Hsu, Investigation, Methodology, Writing – original draft; Andrew Miri, Conceptualization, Investigation, Methodology, Writing – original draft, Writing – review and editing

### Author ORCIDs

Mark Agrios ⬚ https://orcid.org/0000-0002-3792-4843
Feihong Xu ⬚ https://orcid.org/0000-0002-6211-7234
Andrew Miri ⬚ https://orcid.org/0000-0002-5791-7798

### Ethics

All experiments and procedures were performed according to NIH guidelines and approved by the Institutional Animal Care and Use Committee of Northwestern University (protocol IS00009077).

Reviewer #1 (Public review): https://doi.org/10.7554/eLife.103069.3.sa1
Reviewer #2 (Public review): https://doi.org/10.7554/eLife.103069.3.sa2
Reviewer #3 (Public review): https://doi.org/10.7554/eLife.103069.3.sa3
Author response https://doi.org/10.7554/eLife.103069.3.sa4

## Additional files

### Supplementary files

MDAR checklist

### Data availability

The data that support the findings of this study and all Matlab code used for data analyses are available on the Miri lab's GitHub page https://github.com/mirilab-code/forelimbhierarchy (copy archived at *Agrios and mirilab-code, 2025*).

The following dataset was generated:

| Author(s) | Year | Dataset title | Dataset URL | Database and Identifier |
| --- | --- | --- | --- | --- |
| Saiki-Ishikawa A, Agrios M, Savya S, Forrest A, Sroussi H, Hsu S, Basrai D, Xu F, Miri A | 2025 | Mouse forelimb directed cued reaching premotor and primary motor cortex | https://dandiarchive.org/dandiset/001466 | DANDI, 001466 |

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
