## [Editor Report · eLife Assessment]

This study provides **important** insights as to how interacting brain areas produce movement during the execution of a skilled multi-directional reaching task. Using a combination of single neuron and neural population analysis, optogenetic stimulation, and computational models, the authors provide **convincing** evidence of an asymmetrical influence between mouse premotor and motor cortex during the execution of a well-practiced behaviour. This asymmetry can only be captured by some but not all population analysis methods, which is a key lesson to the field in and of itself. Analyzing how activity that is shared and private to these areas relates to different aspects of movements, and why different methods provide different outcomes regarding the nature of inter-area interactions would further strengthen this work.

---

## [Referee Report · Reviewer #1 (Public review)]

This study examined the interaction between two key cortical regions in the mouse brain involved in goal-directed movements, the rostral forelimb area (RFA) - considered a premotor region involved in movement planning, and the caudal forelimb area (CFA) - considered a primary motor region that more directly influences movement execution. The authors ask whether there exists a hierarchical interaction between these regions, as previously hypothesized, and focus on a specific definition of hierarchy - examining whether the neural activity in the premotor region exerts a larger functional influence on the activity in the primary motor area, than vice versa. They examine this question using advanced experimental and analytical methods, including localized optogenetic manipulation of neural activity in either region while measuring both the neural activity in the other region and EMG signals from several muscles involved in the reaching movement, as well as simultaneous electrophysiology recordings from both regions in a separate cohort of animals.

The findings presented show that localized optogenetic manipulation of neural activity in either RFA or CFA resulted in similarly short-latency changes of the muscle output and in firing rate changes in the other region. However, perturbation of RFA led to a larger absolute change in the neural activity of CFA neurons. The authors interpret these findings as evidence for reciprocal, but asymmetrical, influence between the regions, suggesting some degree of hierarchy in which RFA has a greater effect on the neural activity in CFA. They go on to examine whether this asymmetry can also be observed in simultaneously recorded neural activity patterns from both regions. They use multiple advanced analysis methods that either identify latent components in the population level or measure the predictability of firing rates of single neurons in one region using firing rates of single neurons in the other region. Interestingly, the main finding across these analyses seems to be that both regions share highly similar components that capture a high degree of the variability of the neural activity patterns in each region. Single units' activity from either region could be predicted to a similar degree from the activity of single units in the other region, without a clear division into a leading area and a lagging area, as one might expect to find in a simple hierarchical interaction. However, the authors find some evidence showing a slight bias towards leading activity in RFA. Using a two-region neural network model that is fit to the summed neural activity recorded in the different experiments and to the summed muscle output, the authors show that a network with constrained (balanced) weights between the regions can still output the observed measured activities and the observed asymmetrical effects of the optogenetic manipulations, by having different within-region local weights. These results emphasize the challenges in studying interactions between brain regions with reciprocal interactions, multiple external inputs, and recurrent within-region connections.

Strengths:

The experiments and analyses performed in this study are comprehensive and provide a detailed examination and comparison of neural activity recorded simultaneously using dense electrophysiology probes from two main motor regions that have been the focus of studies examining goal-directed movements. The findings showing reciprocal effects from each region to the other, similar short-latency modulation of muscle output by both regions, and similarity of neural activity patterns, are convincing and add to the growing body of evidence that highlight the complexity of the interactions between multiple regions in the motor system and go against a simple feedforward-like hierarchy.

The neural network model complements these findings and adds an important demonstration that the observed asymmetry can, in theory, also arise from differences in local recurrent connections and not necessarily from different input projections from one region to the other. This sheds an important light on the multiple factors that should be considered when studying the interaction between any two brain regions, with a specific emphasis on the role of local recurrent connections, that should be of interest to the general neuroscience community.

Weaknesses:

While the reciprocal interaction and similarity in neural activity across RFA and CFA is an important observation that is supported by the authors' findings, the evidence for a hierarchical interaction between the two regions appears to be weaker. The primary evidence for a hierarchical interaction comes from a causal optogenetic manipulation, carried out at the onset of the reaching movement and conducted with n = 3 in each experimental group, which shows an effect in both regions, yet the effect is greater when silencing the activity in RFA and examining the resulting change in CFA, than vice versa. Analysis of the simultaneously recorded neural activity, on the other hand, reveals mostly no clear hierarchy with leading or lagging dynamics between the regions. The findings of the optogenetic manipulation might be more compelling if similar effects were observed when the same manipulation was applied at different stages of movement preparation and execution, indicating a consistent interaction that is independent from the movement phase.

The methods used to investigate hierarchical interactions through analysis of simultaneously recorded activity yielded inconsistent results. For instance, CCA and PLS showed no clear lead-lag relationship, while DLAG provided some evidence suggesting RFA leads CFA. Overall, these methods largely failed to demonstrate a clear hierarchical interaction. Assuming a partial hierarchy exists, this inconsistency may indicate that the hierarchy is not reflected in the activity patterns or that these analytical methods are inadequate for detecting such interactions within complex neural networks that are influenced by multiple external inputs, reciprocal inter-regional connections, and dominant intra-regional recurrent activity.

As is also argued by the authors, these inconsistent findings underscore the need for caution when interpreting results from similar analyses used to infer inter-regional interactions from neural activity patterns alone. However, the study lacks sufficient explanation for why different methods yielded different results and more elaborate clarification is needed for the findings presented. For example, in the population-level analyses using CCA and PLS, the authors show that both techniques reveal components that are highly similar across regions and explain a substantial portion of each region's variance. Yet, shifting the activity of one region relative to the other to explore potential lead-lag relationships does not alter the results of these analyses. If the regions' activities were better aligned at some unknown true lead-lag time (or aligned at zero), one would expect a peak in alignment within the tested range, as is observed when these same analyses are applied to activity within a single region. It is thus unclear why shifting one region's activity relative to the other does not change the outcome. The interpretation of these results therefore, remains ambiguous and would benefit from further clarification.

---

## [Referee Report · Reviewer #2 (Public review)]

Summary:

While technical advances have enabled large-scale, multi-site neural recordings, characterizing inter-regional communication and its behavioral relevance remains challenging due to intrinsic properties of the brain such as shared inputs, network complexity, and external noise. This work by Saiki-Ishkawa et al. examines the functional hierarchy between premotor (PM) and primary motor (M1) cortices in mice during a directional reaching task. The authors find some evidence consistent with an asymmetric reciprocal influence between the regions, but overall, activity patterns were highly similar and equally predictive of one another. These results suggest that motor cortical hierarchy, though present, is not fully reflected in firing patterns alone.

Strengths:

Inferring functional hierarchies between brain regions, given the complexity of reciprocal and local connectivity, dynamic interactions, and the influence of both shared and independent external inputs, is a challenging task. It requires careful analysis of simultaneous recording data, combined with cross-validation across multiple metrics, to accurately assess the functional relationships between regions. The authors have generated a valuable dataset simultaneously recording from both regions at scale from mice performing a cortex-dependent directional reaching task.

Using electrophysiological and silencing data, the authors found evidence supporting the traditionally assumed asymmetric influence from PM to M1. While earlier studies inferred a functional hierarchy based on partial temporal relationships in firing patterns, the authors applied a series of complementary analyses to rigorously test this hierarchy at both individual neuron and population levels, with robust statistical validation of significance.

In addition, recording combined with brief optogenetic silencing of the other region allowed authors to infer the asymmetric functional influence in a more causal manner. This experiment is well designed to focus on the effect of inactivation manifesting through oligosynaptic connections to support the existence of a premotor to primary motor functional hierarchy.

Subsequent analyses revealed a more complex picture. CCA, PLS, and three measures of predictivity (Granger causality, transfer entropy, and convergent cross mapping) emphasized similarities in firing patterns and cross-region predictability. However, DLAG suggested an imbalance, with RFA capturing CFA variance at a negative time lag, indicating that RFA 'leads' CFA. Taken together these results provide useful insights for current studies of functional hierarchy about potential limitations in inferring hierarchy solely based on firing rates.

While I would detail some questions and issues on specifics of data analyses and modeling below, I appreciate the authors' effort in training RNNs that match some behavioral and recorded neural activity patterns including the inactivation result. The authors point out two components that can determine the across-region influence - (1) the amount of inputs received and (2) the dependence on across-region input, i.e., relative importance of local dynamics, providing useful insights in inferring functional relationships across regions.

Weaknesses:

(1) Trial-averaging was applied in CCA and PLS analyses. While trial-averaging can be appropriate in certain cases, it leads to the loss of trial-to-trial variance, potentially inflating the perceived similarities between the activity in the two regions (Figure 4). Do authors observe comparable degrees of similarity, e.g., variance explained by canonical variables? Also, the authors report conflicting findings regarding the temporal relationship between RFA and CFA when using CCA/PLS versus DLAG. Could this discrepancy be due to the use of trial-averaging in former analyses but not in the latter?

(2) A key strength of the current study is the precise tracking of forelimb muscle activity during a complex motor task involving reaching for four different targets. This rich behavioral data is rarely collected in mice and offers a valuable opportunity to investigate the behavioral relevance of the PM-M1 functional interaction, yet little has been done to explore this aspect in depth. For example, single-trial time courses of inter-regional latent variables acquired from DLAG analysis can be correlated with single-trial muscle activity and/or reach trajectories to examine the behavioral relevance of inter-regional dynamics. Namely, can trial-by-trial change in inter-regional dynamics explain behavioral variability across trials and/or targets? Does the inter-areal interaction change in error trials? Furthermore, the authors could quantify the relative contribution of across-area versus within area dynamics to behavioral variability. It would also be interesting to assess the degree to which across-area and within-area dynamics are correlated. Specifically, can across-area dynamics vary independently from within-area dynamics across trials, potentially operating through a distinct communication subspace?

(3) While network modeling of RFA and CFA activity captured some aspects of behavioral and neural data, I wonder if certain findings such as the connection weight distribution (Figure 7C), across-region input (Figure 7F), and the within-region weights (Figure 7G), primarily resulted from fitting the different overall firing rates between the two regions with CFA exhibiting higher average firing rates. Did the authors account for this firing rate disparity when training the RNNs?

(4) Another way to assess the functional hierarchy is by comparing the time courses of movement representation between the two regions. For example, a linear decoder could be used to compare the amount of information about muscle activity and/or target location as well as time courses thereof between the two regions. This approach is advantageous because it incorporates behavior rather than focusing solely on neural activity. Since one of the main claims of this study is the limitation of inferring functional hierarchy from firing rate data alone, the authors should use the behavior as a lens for examining inter-areal interactions.

Comments on revisions:

I appreciate the authors' thoughtful revisions in response to prior reviews, which I believe have substantially improved the manuscript. In particular, I found the addition of the new section "Manifestations of hierarchy in firing patterns" to be valuable, as it begins to address some of the more complex and potentially conflicting observations

---

## [Referee Report · Reviewer #3 (Public review)]

This study investigates how two cortical regions which are central to the study of rodent motor control (rostral forelimb area, RFA, and caudal forelimb area, CFA) interact during directional forelimb reaching in mice. The authors investigate this interaction using (1) optogenetic manipulations in one area while recording extracellularly from the other, (2) statistical analyses of simultaneous CFA/RFA extracellular recordings, and (3) network modeling. The authors provide solid evidence that asymmetry between RFA and CFA can be observed, although such asymmetry is only observed in certain experimental and analytical contexts.

The authors find asymmetry when applying optogenetic perturbations, reporting a greater impact of RFA inactivation on CFA activity than vice-versa. The authors then investigate asymmetry in endogenous activity during forelimb movements and find asymmetry with some analytical methods but not others. Asymmetry was observed in the onset timing of movement-related deviations of local latent components with RFA leading CFA (computed with PCA) and in a relatively higher proportion and importance of cross-area latent components with RFA leading than CFA leading (computed with DLAG). However, no asymmetry was observed using several other methods that compute cross-area latent dynamics, nor with methods computed on individual neuron pairs across regions. The authors follow up this experimental work by developing a two-area model with asymmetric dependence on cross-area input. This model is used to show that differences in local connectivity can drive asymmetry between two areas with equal amounts of across-region input.

Overall, this work provides a useful demonstration that different cross-area analysis methods result in different conclusions regarding asymmetric interactions between brain areas and suggests careful consideration of methods when analyzing such networks is critical. A deeper examination of why different analytical methods result in observed asymmetry or no asymmetry, analyses that specifically examine neural dynamics informative about details of the movement, or a biological investigation of the hypothesis provided by the model would provide greater clarity regarding the interaction between RFA and CFA.

Strengths:

The authors are rigorous in their experimental and analytical methods, carefully monitoring the impact of their perturbations with simultaneous recordings and providing valid controls for their analytical methods. They cite relevant previous literature that largely agrees with the current work, highlighting the continued ambiguity regarding the extent to which there exists an asymmetry in endogenous activity between RFA and CFA.

A strength of the paper is the evidence for asymmetry provided by optogenetic manipulation. They show that RFA inactivation causes a greater absolute difference in muscle activity than CFA interaction (deviations begin 25-50 ms after laser onset, Figure 1) and that RFA inactivation causes a relatively larger decrease in CFA firing rate than CFA inactivation causes in RFA (deviations begin <25ms after laser onset, Figure 3). The timescales of these changes provide solid evidence for an asymmetry in impact of inactivating RFA/CFA on the other region that could not be driven by differences in feedback from disrupted movement (which would appear with a ~50ms delay).

The authors also utilize a range of different analytical methods, showing an interesting difference between some population-based methods (PCA, DLAG) that observe asymmetry, and single neuron pair methods (granger causality, transfer entropy, and convergent cross mapping) that do not. Moreover, the modeling work presents an interesting potential cause of "hierarchy" or "asymmetry" between brain areas: local connectivity that impacts dependence on across-region input, rather than the amount of across-region input actually present.

Weaknesses:

There is no attempt to examine neural dynamics that are specifically relevant/informative about the details of the ongoing forelimb movement (e.g., kinematics, reach direction). Thus, it may be preemptive to claim that firing patterns alone do not reflect functional influence between RFA/CFA. For example, given evidence that the largest component of motor cortical activity doesn't reflect details of ongoing movement (reach direction or path; Kaufman, et al. PMID: 27761519) and that the analytical tools the authors use likely include this component (PCA, CCA), it may not be surprising that CFA and RFA do not show asymmetry if such asymmetry is related to control of movement details. An asymmetry may still exist in the components of neural activity that encode information about movement details, and thus it may be necessary to isolate and examine the interaction of behaviorally-relevant dynamics (e.g., Sani, et al. PMID: 33169030).

The idea that local circuit dynamics play a central role in determining the asymmetry between RFA and CFA is not supported by experimental data in this paper. The plausibility of this hypothesis is supported by the model but is not explored in any analyses of the experimental data collected. Further experimental investigation is needed to separate this hypothesis from other possibilities.

Comments on revisions:

The authors have improved the manuscript by reviewing several aspects of the text and the addition of supplemental materials. I believe these revisions have clarified some important aspects of the results.

---

## [Author Response]

The following is the authors’ response to the original reviews

**Public Reviews:**

**Reviewer #1 (Public review):**
This study examined the interaction between two key cortical regions in the mouse brain involved in goal-directed movements, the rostral forelimb area (RFA) - considered a premotor region involved in movement planning, and the caudal forelimb area (CFA) - considered a primary motor region that more directly influences movement execution. The authors ask whether there exists a hierarchical interaction between these regions, as previously hypothesized, and focus on a specific definition of hierarchy - examining whether the neural activity in the premotor region exerts a larger functional influence on the activity in the primary motor area than vice versa. They examine this question using advanced experimental and analytical methods, including localized optogenetic manipulation of neural activity in either region while measuring both the neural activity in the other region and EMG signals from several muscles involved in the reaching movement, as well as simultaneous electrophysiology recordings from both regions in a separate cohort of animals.The findings presented show that localized optogenetic manipulation of neural activity in either RFA or CFA resulted in similarly short-latency changes in the muscle output and in firing rate changes in the other region. However, perturbation of RFA led to a larger absolute change in the neural activity of CFA neurons. The authors interpret these findings as evidence for reciprocal, but asymmetrical, influence between the regions, suggesting some degree of hierarchy in which RFA has a greater effect on the neural activity in CFA. They go on to examine whether this asymmetry can also be observed in simultaneously recorded neural activity patterns from both regions. They use multiple advanced analysis methods that either identify latent components at the population level or measure the predictability of firing rates of single neurons in one region using firing rates of single neurons in the other region. Interestingly, the main finding across these analyses seems to be that both regions share highly similar components that capture a high degree of variability of the neural activity patterns in each region. Single units' activity from either region could be predicted to a similar degree from the activity of single units in the other region, without a clear division into a leading area and a lagging area, as one might expect to find in a simple hierarchical interaction. However, the authors find some evidence showing a slight bias towards leading activity in RFA. Using a two-region neural network model that is fit to the summed neural activity recorded in the different experiments and to the summed muscle output, the authors show that a network with constrained (balanced) weights between the regions can still output the observed measured activities and the observed asymmetrical effects of the optogenetic manipulations, by having different within-region local weights. These results put into question whether previous and current findings that demonstrate asymmetry in the output of regions can be interpreted as evidence for asymmetrical (and thus hierarchical) inputs between regions, emphasizing the challenges in studying interactions between any brain regions.Strengths:The experiments and analyses performed in this study are comprehensive and provide a detailed examination and comparison of neural activity recorded simultaneously using dense electrophysiology probes from two main motor regions that have been the focus of studies examining goal-directed movements. The findings showing reciprocal effects from each region to the other, similar short-latency modulation of muscle output by both regions, and similarity of neural activity patterns without a clear lead/lag interaction, are convincing and add to the growing body of evidence that highlight the complexity of the interactions between multiple regions in the motor system and go against a simple feedforward-like network and dynamics. The neural network model complements these findings and adds an important demonstration that the observed asymmetry can, in theory, also arise from differences in local recurrent connections and not necessarily from different input projections from one region to the other. This sheds an important light on the multiple factors that should be considered when studying the interaction between any two brain regions, with a specific emphasis on the role of local recurrent connections, that should be of interest to the general neuroscience community.Weaknesses:While the similarity of the activity patterns across regions and lack of a clear leading/lagging interaction are interesting observations that are mostly supported by the findings presented (however, see comment below for lack of clarity in CCA/PLS analyses), the main question posed by the authors - whether there exists an endogenous hierarchical interaction between RFA and CFA - seems to be left largely open.The authors note that there is currently no clear evidence of asymmetrical reciprocal influence between naturally occurring neural activity patterns of the two regions, as previous attempts have used non-natural electrical stimulation, lesions, or pharmacological inactivation. The use of acute optogenetic perturbations does not seem to be vastly different in that aspect, as it is a non-natural stimulation of inhibitory interneurons that abruptly perturbs the ongoing dynamics.

We do believe that our optogenetic inactivation identifies a causal interaction between the endogenous activity patterns in the excitatory projection neurons, which we have largely silenced, and the downstream endogenous activity that is perturbed. The effect in the downstream region results directly from the silencing of activity in the excitatory projection neurons that mediate each region’s interaction with other regions. Here we have performed a causal intervention common in biology: a loss-of-function experiment. Such experiments generally reveal that a causal interaction of some sort is present, but often do not clarify much about the nature of the interaction, as is true in our case. By showing that a silencing of endogenous activity in one motor cortical region causes a significant change to the endogenous activity in another, we establish a causal relationship between these activity patterns. This is analogous to knocking out the gene for a transcription factor and observing causal effects on the expression of other genes that depend on it.

Moreover, our experiments are, to our knowledge, the first that localize a causal relationship to endogenous activity in motor cortex at a particular point during a motor behavior. Lesion and pharmacological or chemogenetic inactivation have long-lasting effects, and so their consequences on firing in other regions cannot be attributed to a short-latency influence of activity at a particular point during movement. Moreover, the involvement of motor cortex in motor learning and movement preparation/initiation complicates the interpretation of these consequences in relation to movement execution, as disturbance to processes on which execution depends can impede execution itself. Stimulation experiments generate spiking in excitatory projection neurons that is not endogenous.

That said, we would agree that the form of the causal interaction between RFA and CFA remains unaddressed by our results. These results do not expose how the silenced activity patterns affect activity in the downstream region, just as knocking out a transcription factor gene does not expose how the transcription factor influences the expression of other genes. To show evidence for a specific type of interaction dynamics between RFA and CFA, a different sort of experiment would be necessary. See Jazayeri and Afraz, Neuron, 2017 for more on this issue.

Furthermore, the main finding that supports a hierarchical interaction is a difference in the absolute change of firing rates as a result of the optogenetic perturbation, a finding that is based on a small number of animals (N = 3 in each experimental group), and one which may be difficult to interpret.

Though N = 3, we do show statistical significance. Moreover, using three replicates is not uncommon in biological experiments that require a large technical investment.

As the authors nicely demonstrate in their neural network model, the two regions may differ in the strength of local within-region inhibitory connections. Could this theoretically also lead to a difference in the effect of the artificial light stimulation of the inhibitory interneurons on the local population of excitatory projection neurons, driving an asymmetrical effect on the downstream region?

We (Miri et al., Neuron, 2017) and others (Guo et al., Neuron, 2014) have shown that the effect of this inactivation on excitatory neurons in CFA is a near-complete silencing (90-95% within 20 ms). There thus is not much room for the effects on projection neurons in RFA to be much larger. We have measured these local effects in RFA as part of other work (Kristl et al., biorxiv, 2025), verifying that the effects on RFA projection neuron firing are not larger.

Moreover, the manipulation was performed upon the beginning of the reaching movement, while the premotor region is often hypothesized to exert its main control during movement preparation, and thus possibly show greater modulation during that movement epoch. It is not clear if the observed difference in absolute change is dependent on the chosen time of optogenetic stimulation and if this effect is a general effect that will hold if the stimulation is delivered during different movement epochs, such as during movement preparation.

We agree that the dependence of RFA-CFA interactions on movement phase would be interesting to address in subsequent experiments. While a strong interpretation of lesion results might lead to a hypothesis that premotor influence on primary motor cortex is local to, or stronger during, movement preparation as opposed to execution, at present there is to our knowledge no empirical support from interventional experiments for this hypothesis. Moreover, existing results from analysis of activity in these two regions have produced conflicting results on the strength of interaction between these regions during preparation. Compare for example BachschmidRomano et al., eLife, 2023 to Kaufman et al., Nature Neuroscience, 2014.

That said, this lesion interpretation would predict the same asymmetry we have observed from perturbations at the beginning of a reach - a larger effect of RFA on CFA than vice versa.

Another finding that is not clearly interpretable is in the analysis of the population activity using CCA and PLS. The authors show that shifting the activity of one region compared to the other, in an attempt to find the optimal leading/lagging interaction, does not affect the results of these analyses. Assuming the activities of both regions are better aligned at some unknown groundtruth lead/lag time, I would expect to see a peak somewhere in the range examined, as is nicely shown when running the same analyses on a single region's activity. If the activities are indeed aligned at zero, without a clear leading/lagging interaction, but the results remain similar when shifting the activities of one region compared to the other, the interpretation of these analyses is not clear.

Our results in this case were definitely surprising. Many share the intuition that there should be a lag at which the correlations in activity between regions may be strongest. The similarity in alignment across lags we observed might be expected if communication between regions occurs over a range of latencies as a result of dependence on a broad diversity of synaptic paths that connect neurons. In the Discussion, we offer an explanation of how to reconcile these findings with the seemingly different picture presented by DLAG.

**Reviewer #2 (Public review):**
Summary:While technical advances have enabled large-scale, multi-site neural recordings, characterizing inter-regional communication and its behavioral relevance remains challenging due to intrinsic properties of the brain such as shared inputs, network complexity, and external noise. This work by Saiki-Ishkawa et al. examines the functional hierarchy between premotor (PM) and primary motor (M1) cortices in mice during a directional reaching task. The authors find some evidence consistent with an asymmetric reciprocal influence between the regions, but overall, activity patterns were highly similar and equally predictive of one another. These results suggest that motor cortical hierarchy, though present, is not fully reflected in firing patterns alone.Strengths:Inferring functional hierarchies between brain regions, given the complexity of reciprocal and local connectivity, dynamic interactions, and the influence of both shared and independent external inputs, is a challenging task. It requires careful analysis of simultaneous recording data, combined with cross-validation across multiple metrics, to accurately assess the functional relationships between regions. The authors have generated a valuable dataset simultaneously recording from both regions at scale from mice performing a cortex-dependent directional reaching task.Using electrophysiological and silencing data, the authors found evidence supporting the traditionally assumed asymmetric influence from PM to M1. While earlier studies inferred a functional hierarchy based on partial temporal relationships in firing patterns, the authors applied a series of complementary analyses to rigorously test this hierarchy at both individual neuron and population levels, with robust statistical validation of significance.In addition, recording combined with brief optogenetic silencing of the other region allowed authors to infer the asymmetric functional influence in a more causal manner. This experiment is well designed to focus on the effect of inactivation manifesting through oligosynaptic connections to support the existence of a premotor to primary motor functional hierarchy.Subsequent analyses revealed a more complex picture. CCA, PLS, and three measures of predictivity (Granger causality, transfer entropy, and convergent cross-mapping) emphasized similarities in firing patterns and cross-region predictability. However, DLAG suggested an imbalance, with RFA capturing CFA variance at a negative time lag, indicating that RFA 'leads' CFA. Taken together these results provide useful insights for current studies of functional hierarchy about potential limitations in inferring hierarchy solely based on firing rates.While I would detail some questions and issues on specifics of data analyses and modeling below, I appreciate the authors' effort in training RNNs that match some behavioral and recorded neural activity patterns including the inactivation result. The authors point out two components that can determine the across-region influence - (1) the amount of inputs received and (2) the dependence on across-region input, i.e., the relative importance of local dynamics, providing useful insights in inferring functional relationships across regions.Weaknesses:(1) Trial-averaging was applied in CCA and PLS analyses. While trial-averaging can be appropriate in certain cases, it leads to the loss of trial-to-trial variance, potentially inflating the perceived similarities between the activity in the two regions (Figure 4). Do authors observe comparable degrees of similarity, e.g., variance explained by canonical variables? Also, the authors report conflicting findings regarding the temporal relationship between RFA and CFA when using CCA/PLS versus DLAG. Could this discrepancy be due to the use of trial-averaging in former analyses but not in the latter?

We certainly agree that the similarity in firing patterns is higher in trial averages than on single trials, given the variation in single-neuron firing patterns across trials. Here, we were trying to examine the similarity of activity variance that is clearly movement dependent, as trial averages are, and to use an approach aligned with those applied in the existing literature. We would also agree that there is more that can be learned about interactions from trial-by-trial analysis. It is possible that the activity components identified by DLAG as being asymmetric somehow are not reflected strongly in trial averages. In our Discussion we offer another potential explanation that is based on other differences in what is calculated by DLAG and CCA/PLS.

We also note here that all of the firing pattern predictivity analysis we report (Figure 6) was done on single trial data, and in all cases the predictivity was symmetric. Thus, our results in aggregate are not consistent with symmetry purely being an artifact of trial averaging.

(2) A key strength of the current study is the precise tracking of forelimb muscle activity during a complex motor task involving reaching for four different targets. This rich behavioral data is rarely collected in mice and offers a valuable opportunity to investigate the behavioral relevance of the PM-M1 functional interaction, yet little has been done to explore this aspect in depth. For example, single-trial time courses of inter-regional latent variables acquired from DLAG analysis can be correlated with single-trial muscle activity and/or reach trajectories to examine the behavioral relevance of inter-regional dynamics. Namely, can trial-by-trial change in inter-regional dynamics explain behavioral variability across trials and/or targets? Does the inter-areal interaction change in error trials? Furthermore, the authors could quantify the relative contribution of across-area versus within-area dynamics to behavioral variability. It would also be interesting to assess the degree to which across-area and within-area dynamics are correlated. Specifically, can acrossarea dynamics vary independently from within-area dynamics across trials, potentially operating through a distinct communication subspace?

These are all very interesting questions. Our study does not attempt to parse activity into components predictive of muscle activity and others that may reflect other functions. Distinct components of RFA and CFA activity may very well rely on distinct interactions between them.

(3) While network modeling of RFA and CFA activity captured some aspects of behavioral and neural data, I wonder if certain findings such as the connection weight distribution (Figure 7C), across-region input (Figure 7F), and the within-region weights (Figure 7G), primarily resulted from fitting the different overall firing rates between the two regions with CFA exhibiting higher average firing rates. Did the authors account for this firing rate disparity when training the RNNs?

The key comparison in Figure 7 is shown in 7F, where the firing rates are accounted for in calculating the across-region input strength. Equalizing the firing rates in RFA and CFA would effectively increase RFA rates. If the mean firing rates in each region were appreciably dependent on across-region inputs, we would then expect an off-setting change in the RFA→CFA weights, such that the RFA→CFA distributions in 7F would stay the same. We would also expect the CFA→RFA weights would increase, since RFA neurons would need more input. This would shift the CFA→RFA (blue) distributions up. Thus, if anything, the key difference in this panel would only get larger.

We also generally feel that it is a better approach to fit the actual firing rates, rather than normalizing, since normalizing the firing rates would take us further from the actual biology, not closer.

(4) Another way to assess the functional hierarchy is by comparing the time courses of movement representation between the two regions. For example, a linear decoder could be used to compare the amount of information about muscle activity and/or target location as well as time courses thereof between the two regions. This approach is advantageous because it incorporates behavior rather than focusing solely on neural activity. Since one of the main claims of this study is the limitation of inferring functional hierarchy from firing rate data alone, the authors should use the behavior as a lens for examining inter-areal interactions.

As we state above, we agree that examining interactions specific to movement-related activity components could reveal interesting structure in interregional interactions. Since it remains a challenge to rigorously identify a subset of neural activity patterns specifically related to driving muscle activity, any such analysis would involve an additional assumption. It remains unclear how well the activity that decoders use for predicting muscle activity matches the activity that actually drives muscle activity in situ.

To address this issue, which related to one raised by Reviewer #3 below, we have added an additional paragraph to the Discussion (see “Manifestations of hierarchy in firing patterns”).

**Reviewer #3 (Public review):**
This study investigates how two cortical regions that are central to the study of rodent motor control (rostral forelimb area, RFA, and caudal forelimb area, CFA) interact during directional forelimb reaching in mice. The authors investigate this interaction using(1) optogenetic manipulations in one area while recording extracellularly from the other, (2) statistical analyses of simultaneous CFA/RFA extracellular recordings, and (3) network modeling.The authors provide solid evidence that asymmetry between RFA and CFA can be observed, although such asymmetry is only observed in certain experimental and analytical contexts.The authors find asymmetry when applying optogenetic perturbations, reporting a greater impact of RFA inactivation on CFA activity than vice-versa. The authors then investigate asymmetry in endogenous activity during forelimb movements and find asymmetry with some analytical methods but not others. Asymmetry was observed in the onset timing of movement-related deviations of local latent components with RFA leading CFA (computed with PCA) and in a relatively higher proportion and importance of cross-area latent components with RFA leading than CFA leading (computed with DLAG). However, no asymmetry was observed using several other methods that compute cross-area latent dynamics, nor with methods computed on individual neuron pairs across regions. The authors follow up this experimental work by developing a twoarea model with asymmetric dependence on cross-area input. This model is used to show that differences in local connectivity can drive asymmetry between two areas with equal amounts of across-region input.Overall, this work provides a useful demonstration that different cross-area analysis methods result in different conclusions regarding asymmetric interactions between brain areas and suggests careful consideration of methods when analyzing such networks is critical. A deeper examination of why different analytical methods result in observed asymmetry or no asymmetry, analyses that specifically examine neural dynamics informative about details of the movement, or a biological investigation of the hypothesis provided by the model would provide greater clarity regarding the interaction between RFA and CFA.Strengths:The authors are rigorous in their experimental and analytical methods, carefully monitoring the impact of their perturbations with simultaneous recordings, and providing valid controls for their analytical methods. They cite relevant previous literature that largely agrees with the current work, highlighting the continued ambiguity regarding the extent to which there exists an asymmetry in endogenous activity between RFA and CFA.A strength of the paper is the evidence for asymmetry provided by optogenetic manipulation. They show that RFA inactivation causes a greater absolute difference in muscle activity than CFA interaction (deviations begin 25-50 ms after laser onset, Figure 1) and that RFA inactivation causes a relatively larger decrease in CFA firing rate than CFA inactivation causes in RFA (deviations begin <25ms after laser onset, Figure 3). The timescales of these changes provide solid evidence for an asymmetry in the impact of inactivating RFA/CFA on the other region that could not be driven by differences in feedback from disrupted movement (which would appear with a ~50ms delay).The authors also utilize a range of different analytical methods, showing an interesting difference between some population-based methods (PCA, DLAG) that observe asymmetry, and single neuron pair methods (granger causality, transfer entropy, and convergent cross mapping) that do not. Moreover, the modeling work presents an interesting potential cause of "hierarchy" or "asymmetry" between brain areas: local connectivity that impacts dependence on across-region input, rather than the amount of across-region input actually present.Weaknesses:There is no attempt to examine neural dynamics that are specifically relevant/informative about the details of the ongoing forelimb movement (e.g., kinematics, reach direction). Thus, it may be preemptive to claim that firing patterns alone do not reflect functional influence between RFA/CFA. For example, given evidence that the largest component of motor cortical activity doesn't reflect details of ongoing movement (reach direction or path; Kaufman, et al. PMID: 27761519) and that the analytical tools the authors use likely isolate this component (PCA, CCA), it may not be surprising that CFA and RFA do not show asymmetry if such asymmetry is related to the control of movement details.An asymmetry may still exist in the components of neural activity that encode information about movement details, and thus it may be necessary to isolate and examine the interaction of behaviorally-relevant dynamics (e.g., Sani, et al. PMID: 33169030).

To clarify, we are not claiming that firing patterns in no way reflect the asymmetric functional influence that we demonstrate with optogenetic inactivation. Instead, we show that certain types of analysis that we might expect to reflect such influence, in fact, do not. Indeed, DLAG did exhibit asymmetries that matched those seen in functional influence (at least qualitatively), though other methods we applied did not.

As we state above, we do think that there is more that can be gleaned by looking at influence specifically in terms of activity related to movement. However, if we did find that movement-related activity exhibited an asymmetry following functional influence, our results imply that the remaining activity components would exhibit an opposite asymmetry, such that the overall balance is symmetric. This would itself be surprising. We also note that the components identified by CCA and PLS do show substantial variation across reach targets, indicating that they are not only reflecting condition-invariant components. These analyses were performed on components accounting for well over 90% of the total activity variance, suggesting that both conditiondependent and condition-invariant components should be included.

To address the concern about condition-dependent and condition-invariant components, we have added a sentence to the Results section reporting our CCA and PLS results: “Because our results here involve the vast majority of trial-averaged activity variance, we expect that they encompass both components of activity that vary for different movement conditions (condition-dependent), and those that do not (condition-invariant).” To address the general concerns about potential differences in activity components specifically related to muscle activity, we have also added an additional paragraph to the Discussion (see “Manifestations of hierarchy in firing patterns”).

The idea that local circuit dynamics play a central role in determining the asymmetry between RFA and CFA is not supported by experimental data in this paper. The plausibility of this hypothesis is supported by the model but is not explored in any analyses of the experimental data collected. Given the focus on this idea in the discussion, further experimental investigation is warranted.

While we do not provide experimental support for this hypothesis, the data we present also do not contradict this hypothesis. Here we used modeling as it is often used - to capture experimental results and generate hypotheses about potential explanation. We do feel that our Discussion makes clear where the hypothesis derives from and does not misrepresent the lack of experimental support. We expect readers will take our engagement with this hypothesis with the appropriate grain of salt. The imaginable experiments to support such a hypothesis would constitute another substantial study, requiring numerous controls - a whole other paper in itself.

**Recommendations for the authors:**

**Reviewer #1 (Recommendations for the authors):**
(1) There are a few small text/figure caption modifications that can be made for clarity of reading:(2) Unclear sentence in the second paragraph of the introduction: "For example, stimulation applied in PM has been shown to alter the effects on muscles of stimulation in M1 under anesthesia, both in monkeys and rodents."

This sentence has been rephrased for clarity: “For example, in anesthetized monkeys34 and rodents35, stimulation in PM alters the effects of stimulation in M1 on muscles.”

(3) The first section of the results presents the optogenetic manipulation. However, the critical control that tests whether this was strictly a local manipulation that did not affect cells in the other region is introduced only much later. It may be helpful to add a comment in this section noting that such a control was performed, even if it is explained in detail later when introducing the recordings.

We have added the following to the first Results section: “we show below that direct optogenetic effects were only seen in the targeted forelimb area and not the other.”

(4) Figure 1D - I imagine these averages are from a single animal, but this is not stated in the figure caption.

“For one example mouse,” has been added to the beginning of the Figure 1D legend.

(5) Figure 2F - N=6 is not stated in the panel's caption (though it can make it clearer), while it is stated in the caption of 2H.

“n = 6 mice” has been added to the Figure 2F legend.

(6) There's some inconsistency with the order of RFA/CFA in the figures, sometimes RFA is presented first (e.g., Figure 1D and 1F), and sometimes CFA is presented first (e.g., panels of Figure 2).

We do not foresee this leading to confusion.

(7) "As expected, the majority of recorded neurons in each region exhibited an elevated average firing rate during movement as compared to periods when forelimb muscles were quiescent (Figure 2D,E; Figure S1A,B)" - Figure S1A,B show histograms of narrow vs. wide waveforms, is this the relevant figure here?

We apologize for the cryptic reference. The waveform width histograms were referred to here because they enabled the separation of narrow- and wide-waveform cells shown in Figure 2D,E. We have added the following clause to the referenced sentence to make this explicit: “, both for narrow-waveform, putative interneurons and wide-waveform putative pyramidal neurons.”

(8) Figure 2I caption - "The fraction of activity variance from 150 ms before reach onset to 150 ms after it that occurs before reach onset" - this sentence is not clear.

The Figure 2I legend has been updated to “The activity variance in the 150 ms before muscle activity onset, defined as a fraction of the total activity variance from 150 ms before to 150 ms after muscle activity onset, for each animal (circles) and the mean across animals (black bars, n = 6 mice).”

(9) Figure 4B-G - is this showing results across the 6 animals? Not stated clearly.

Yes - the 21 sessions we had referred to are drawn from all six mice. We have updated the legend here to make this explicit.

(10) DLAG analysis - is there any particular reasoning behind choosing four across-region and four within-region components?

In actuality, we completed this analysis for a broad range of component numbers and obtained similar results in all cases. Four fell in the center of our range, and so we focused the illustrations shown in the figure on this value. In general, the number of components is arbitrary. The original paper from Gokcen et al. describes a method for identifying a lower bound on the number of distinct components the method can identify. However, this method yields different results for each individual recording session. For the comparisons we performed, we needed to use the same range of values for each session.

(11) Figure 5A seems to show 11 across-session components, it's unclear from the caption but I imagine this should show 12 (4 components times 3 sessions?)

As we state in the Methods, any across-region latent variable with a lag that failed to converge between the boundary values of ±200 ms was removed from the analysis. In the case illustrated in this panel, the lag for one of the components failed to converge and is not shown. We have now clarified this both in the relevant Results paragraph and in the figure legend.

(12) Figure 5B - is each marker here the average variance explained by all across/within components that were within the specified lag criteria across sessions per mouse? In other words, what does a single marker here stand for?

We apologize for the lack of clarity here. These values reflect the average across sessions for each mouse. We have updated the legend to make this explicit.

**Reviewer #2 (Recommendations for the authors):**
As I have addressed most of my major recommendations in the public review, I will use this section to include relatively minor points for the authors to consider.(1) The EMG data in Figure 1C shows distinct patterns across spouts, both in the magnitude and complexity of muscle activations. It would be interesting to investigate whether these differences in muscle activity lead to behavioral variations (e.g., reaction time, reach duration) and how they relate to the relative involvement of the two areas.

We agree that it would be interesting to examine how the interactions between areas vary as behavior varies. While the differences between reaches here are limited, we have addressed this question for two substantially different motor behaviors (reaching and climbing) in a follow-up study that was recently preprinted (Kristl et al., biorxiv, 2025).

(2) How do the authors account for the lingering impact of RFA inactivation on muscle activity, which persists for tens of milliseconds after laser offset? Could this effect be due to compensatory motor activity following the perturbation? A further illustration of how the raw limb trajectories and/or muscle activity are perturbed and recovered would help readers better understand the impact of motor cortical inactivation.

To clarify the effects of inactivation on a longer timescale, we have added a new supplemental figure showing the plots from Figure 1D over a longer time window extending to 500 ms after trial onset (new Figure S1). Lingering effects do persist, at least in certain cases. In general, we find it hard to ascertain the source of optogenetic effects on longer timescales like this. On the shortest timescales, effects will be mediated by relatively direct connections between regions. However, on these longer timescales, effects could be due to broader changes in brain and behavioral state that can influence muscle activity. For example, attempts to compensate for the initial disturbance to muscle activity could cause divergence from controls on these longer timescales. Muscle tissue itself is also known to have long timescale relaxation dynamics, and it would not be surprising if the relevant control circuits here also had long timescales dynamics, such that we would not expect an immediate return to control when the light pulse ends. Because of this ambiguity, we generally avoid interpretation of optogenetic effects on these longer timescales.

**Reviewer #3 (Recommendations for the authors):**
(1) Page 9: ". We measured the time at which the activity state deviated from baseline preceding reach onset," - I cannot find how this deviation was defined (neither the baseline nor the threshold).

We have added text to the Figure 2G legend that explicitly states how the baseline and activity onset time were defined.

(2) Given the shape of the curves in Figure 2G, the significance of this result seems susceptible to slight modifications of what defines a baseline or a deviation threshold. For example, it looks like the circle for CFA has a higher y-axis value, suggesting the baseline deviance is higher, but it is unclear why that would be from the plot. If the threshold for deviation in neural activity state were held uniform between CFA and RFA is the difference still significant across animals?

We have repeated the analysis using the same absolute threshold for each region. We used the higher of the two thresholds from each region. The difference remains significant. This is now described in the last paragraph of the Results section for Figure 2.

(3) Since summed deviation of the top 3 PCs is used to show a difference in activity onset between CFA/RFA, but only a small proportion of variance is explained pre-movement (<2% in most animals), it seems relevant to understand what percentage of CFA/RFA neuron activity actually is modulated and deviates from baseline prior to movement and to show the distribution of activity onsets at the single neuron level in CFA/RFA. Can an onset difference only be observed using PCA?

Because many neurons have low firing rates, estimating the time at which their firing rate begins to rise near reach onset is difficult to do reliably. It is also true that not all neurons show an increase around onset - some show a decrease and others show no discernible change. Using PCs to measure onset avoids both of these problems, since they capture both increases and decreases in individual neuron firing rates and are much less noisy than individual neuron firing rates.

However, based on this comment, we have repeated this analysis on a single-neuron level using only neurons with relatively high average firing rates. Specifically, we analyzed neurons with mean firing rates above the 90th percentile across all sessions within an animal. Neurons whose activity never crossed threshold were excluded. Results matched those using PCs, with RFA neurons showing an earlier average activity onset time. This is now described in the last paragraph of the Results section for Figure 2.

(4) It is stated that to study the impact of inactivation on CFA/RFA activity, only the 50 highest average firing rate neurons were used (and maybe elsewhere too, e.g., convergent cross mapping). It is unclear why this subselection is necessary. It is justified by stating that higher firing rate neurons have better firing rate estimates. This may be supportable for very low firing rate units that spike sorting tools have a hard time tracking, but I don't think this is supported by data for most of the distribution of firing rates. It therefore seems like the results might be biased by a subselection of certain high firing rate neuron populations. It would be useful to also compute and mention if the results for all neurons/neuron pairs are the same. If there is worry about low-quality units being those with low firing rates, a threshold for firing rate as used elsewhere in the paper (at least 1 spike / 2 trials) seems justified.

The issue here is that as firing rates decrease and firing rate estimates get noisier, estimates of the change in firing rate get more variable. Here we are trying to estimate the fraction of neurons for which firing rates decreased upon inactivation of the other region. Variability in estimates of the firing rate change will bias this estimate toward 50%, since in the limit when the change estimates are entirely based on noise, we expect 50% to be decreases. As expected, when we use increasingly liberal thresholds for this analysis, the fraction of decreases trends closer to 50%.

As a consequence of this, we cannot easily distinguish whether higher firing rate neurons might for some reason have a greater tendency to exhibit decreases in firing compared to lower firing rate neurons. However, we see no positive reason to expect such a difference. We have added a sentence noting this caveat in interpreting our findings to the relevant paragraph of the Results.

The lack of min/max axis values in Figure 3B-F makes it hard to interpret - are these neurons almost silent when near the bottom of the plot or are they still firing a substantial # of spikes?

To aid interpretation of the relative magnitude of firing rate changes, we have added minimum firing rates for the averages depicted in Figure 3B,C,E and F to the legend. Our original thinking was that the plots in Figure 3G and H would provide an indication of the relative changes in firing.

It would be interesting to know if the impact of optogenetic stimulation changed with exposure to the manipulation. Are all results presented only from the first X number of sessions in each animal? Or is the effect robust over time and (within the same animal) you can get the same results of optogenetic inactivation over time? This information seems critical for reproducibility.

We have now performed brief optogenetic inactivations in several brain areas in several different behavioral paradigms, and have found that inactivation effects are stable both within and across sessions, almost surprisingly so. This includes cases where the inactivations were more frequent (every ~1.25 s on average) and more numerous (>15,000 trials per animal) than in the present manuscript. Thus we did not restrict our analysis here to the first X sessions or trials within a session. We have added additional plots as Figure S3T-AA showing the stability of optogenetic effects both within and across sessions.

Given that it can be difficult to record from interneurons (as the proportion of putative interneurons in Figure S1 attests), the SALT analyses would be more convincing if a few recordings had been performed in the same region as optogenetic stimulation to show a "positive control" of what direct interneuron stimulation looks like. Could also use this to validate the narrow/wide waveform classification.

We have verified that using SALT as we have in the present manuscript does detect vGAT+ interneurons directly responding to light. This is included in a recent preprint from the lab (Kristl et al., biorxiv, 2025). We (Warriner et al., Cell Reports, 2022) and others (Guo et al., Neuron, 2014) have previously used direct ChR2 activation to validate waveform-based classification.

Simultaneous CFA/RFA recordings during optogenetic perturbation would also allow for time courses of inhibition to be compared in RFA/CFA. Does it take 25ms to inhibit locally, and the cross-area impact is fast, or does it inactivate very fast locally and takes ~25ms to impact the other region?

Latencies of this sort are difficult to precisely measure given the statistical limits of this sort of data, but there does appear to be some degree of delay between local and downstream effects. We do not have a statistical foundation as of yet for concluding that this is the case. It will be interesting to examine this issue more rigorously in the future.

Given the difference in the analytical methods, the authors should share data in a relatively unprocessed format (e.g., spike times from sorted units relative to video tracking + behavioral data), along with analysis code, to allow others to investigate these differences.

We plan to post the data and code to our lab’s Github site once the Version of Record is online.